# CONFHIT: CONFORMAL GENERATIVE DESIGN WITH ORACLE FREE GUARANTEES

**Siddhartha Laghuvarapu**[1], **Ying Jin**[2][*], **Jimeng Sun**[1][*]

[1] Siebel School of Computing and Data Science, University of Illinois Urbana-Champaign, IL, USA
[2] Department of Statistics and Data Science, Wharton School, University of Pennsylvania, PA, USA
{sl160,jimeng}@illinois.edu, yjinstat@wharton.upenn.edu

## ABSTRACT

The success of deep generative models in scientific discovery requires not only the ability to generate novel candidates but also reliable guarantees that these candidates indeed satisfy desired properties. Recent conformal-prediction methods offer a path to such guarantees, but its application to generative modeling in drug discovery is limited by budget constraints, lack of oracle access, and distribution shift. To address these challenges, we introduce CONFHIT, a model-agnostic framework that provides validity guarantees under these conditions. CONFHIT formalizes two central questions: (i) Certification: whether a generated batch can be guaranteed to contain at least one hit with a user-specified confidence level, and (ii) Design: whether the generation can be refined to a compact set without weakening this guarantee. CONFHIT leverages weighted exchangeability between historical and generated samples to eliminate the need for an experimental oracle, constructs multiple-sample density-ratio weighted conformal p-value to quantify statistical confidence in hits, and proposes a nested testing procedure to certify and refine candidate sets of multiple generated samples while maintaining statistical guarantees. Across representative generative molecule design tasks and a broad range of methods, CONFHIT consistently delivers valid coverage guarantees at multiple confidence levels while maintaining compact certified sets, thereby establishing a principled and reliable framework for generative modeling.

## 1 INTRODUCTION

Deep generative modeling has demonstrated remarkable ability to explore high-dimensional spaces, driving advances in various applications like text generation (Radford et al., 2018), image synthesis (Ho et al., 2022), protein engineering (Madani et al., 2020), and molecular discovery (Gómez-Bombarelli et al., 2018). In critical domains such as drug discovery, however, the success requires more than generation: While powerful generative models have been developed to accelerate early-stage discovery (Madani et al., 2021; Hoogeboom et al., 2022; Yim et al., 2023), their practical utility depends on whether the generated candidates indeed satisfy key biochemical properties. Since these properties can only be verified through costly wet-lab or in-vivo experiments, it is crucial to assess and guarantee, in advance, the viability of the generated samples (hits), leading to a central question:

> *Given a generative model, how to construct batches of generated samples that can,*
> *with high statistical confidence, be guaranteed to contain at least one valid hit?*

Conformal prediction provides a model-agnostic framework for establishing statistical guarantees for black-box prediction models (Vovk et al., 2005), and is recently extended to calibrate any generative model to produce sets of generated samples that contain at least one high-quality instance with high probability (Quach et al., 2023; Ulmer et al., 2024; Shahrokhi et al., 2025). While such guarantees can be profoundly useful, direct application of the existing methods in resource-restricted problems like drug discovery is limited by several challenges. (i) *Certification:* With limited generation budget and no assumptions on the generative model, it is not always feasible to produce a valid hit – therefore important to state clearly when a guarantee can be provided and when it cannot. On the technical side, (ii) *Lack of oracle access*: Existing methods rely on an oracle that evaluates newly generated samples for existing inputs (such as by comparing to a gold-standard output). In drug discovery, this means one needs to synthesize and experimentally validate the generated samples, which is infeasible in resource-limited settings alike (Kladny et al., 2024). (iii) *Distribution shift*: The generated candidates may follow a distribution different from the calibration data, violating the exchangeability assumption.

---

[*]These authors jointly supervised this work.

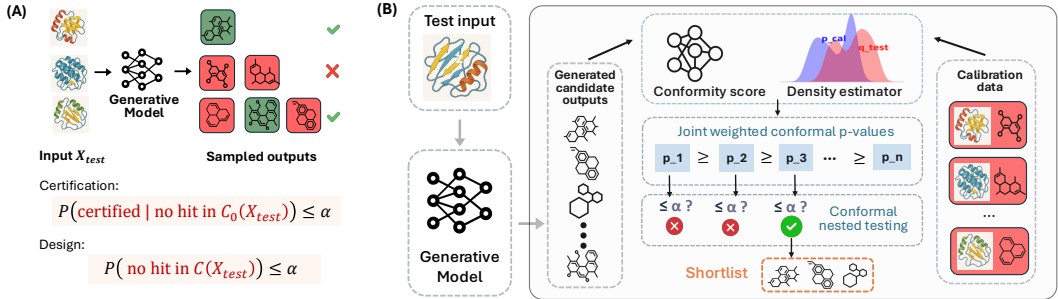

Figure 1: (a) Problem setup: given an input, certify and generate a set of candidates that contains at least one "hit" (green) with probability at least $1 - \alpha$. (b) CONFHIT workflow. Given a nested sequence of candidate batches, we estimate the density ratio between labeled data and generated samples, compute a conformal p-value for each batch to quantify the confidence in it containing a hit, and return the smallest batch whose p-value falls below $\alpha$.

We introduce CONFHIT, a model-agnostic framework that provides reliability guarantees for generative modeling in resource-constrained settings. To tackle the budget limitation, we expand the generation problem into two connected fundamental questions:

1. *Certification.* Given an input $\bar{X}_{\text{test}}$, a batch $\mathcal{C}_0(\bar{X}_{\text{test}})$ of generated samples and an unknown property indicator $A(\cdot) \in \{0, 1\}$, can we guarantee, at a given confidence level $1 - \alpha \in (0, 1)$, that the batch $\mathcal{C}(\bar{X}_{\text{test}})$ contains at least a hit?
2. *Design.* In cases with strong confidence, can we design a compact candidate set $\mathcal{C}(\bar{X}_{\text{test}})$ while preserving the guarantee that it contains a valid hit with probability at least $1 - \alpha$?

Here $A(\cdot) \in \{0, 1\}$ is an oracle that returns 1 only when the sample satisfies a desired property. In molecule design, it may indicate whether a molecule $x$ improves upon a seed molecule $\bar{X}_{\text{test}}$ in terms of activity, and these questions ensure the experimental validation on $\mathcal{C}(\bar{X}_{\text{test}})$ is unlikely to be wasted effort (Figure 1a). We remark that, in resource-abundant settings, certification and refinement can be combined: the generation budget may be enlarged until certification is achieved, after which the set is pruned to a compact certified subset, leading to the same validity guarantees by existing methods.

To eliminate oracle access and handle the distribution shift, CONFHIT leverages the exchangeability structure between historical (labeled) and generated samples, rather than among generated samples for both existing and new model inputs as in earlier methods, and estimates their density ratio, thereby inducing *weighted* exchangeability (Tibshirani et al., 2019; Jin & Candès, 2023a). For the certification problem, given any viability-scoring function, we construct a weighted permutation p-value that quantifies the evidence against the global null that none of the samples in the generated batch are viable. For the design problem, CONFHIT then examines the certification confidence of a nested sequence of candidate batches, and returns the smallest batch that can be certified at a given confidence level $\alpha \in (0, 1)$. We show that this procedure bounds the probability of returning a batch with no viable sample below $\alpha \in (0, 1)$, regardless of the scoring function and the generative model.

We demonstrate the robust guarantees of CONFHIT on two representative generative design tasks: (i) **Constrained molecule optimisation**, which seeks a new molecule that satisfies a target property while remaining similar to a given scaffold; **(ii) Structure-based drug discovery**, which aims to generate molecules that are active against a given protein. We summarize our contributions below:

- We formalise the task of generative modeling in resource-constrained settings with a conformal validity guarantee: given an input context (such as a lead molecule or protein pocket), certify and produce a set of candidates that contains at least one hit at a pre-specified confidence level $1 - \alpha$.

- We introduce a class of density-ratio-weighted, multiple-test-sample conformal p-values for the certification problem, and show their validity for certifying the existence of at least one hit in a given batch of generated samples under distribution shift.

- We propose a general nested testing framework that achieves the validity guarantees for the design problem. In specific, we show that given a sequence of our p-values, stopping as soon as the p-value drops below $\alpha$ achieves finite-sample error control.

- We develop practical strategies for two key elements in the method: score modelling and density ratio estimation, and demonstrate the robust performance on two standard molecule design tasks: constrained molecule optimisation and structure-based drug discovery.

**Related Work.** Conformal prediction (CP) (Vovk et al., 2005) offers distribution-free uncertainty quantification for any black-box model by constructing prediction sets from exchangeable calibration data (Papadopoulos et al., 2002). Recent advances extend CP to generative tasks (Angelopoulos et al., 2021; Quach et al., 2023; Shahrokhi et al., 2025; Kladny et al., 2024). However, these methods require oracle access, which is impractical for resource-constrained scenarios like drug discovery.

In scientific/drug discovery tasks, conformal prediction has so far primarily been used in predictive inference for regression and classification problems in property prediction (Sun et al., 2017; Svensson et al., 2018; Cortés-Ciriano & Bender, 2018; 2019; Zhang et al., 2021) and their extension to covariate-shift settings via importance-weight estimation (Fannjiang et al., 2022; Laghuvarapu et al., 2023; Prinster et al., 2023; Fannjiang & Park, 2025). Beyond prediction-set construction, conformal selection employs p-values to prioritize compounds with selective error control (Bai et al., 2024; Jin & Candès, 2023b; Bai & Jin, 2024). As we address a distinct generative design problem, our formulation, inference scheme, and methodology differ sharply from prior work.

Meanwhile, there is a growing literature on generative modelling in scientific design tasks, including goal-directed small-molecule optimization (Gómez-Bombarelli et al., 2018; Jin et al., 2020), structure-based ligand design (Corso et al., 2022; Guan et al., 2023), large-scale protein generators (Madani et al., 2020; Ingraham et al., 2019; Strokach & Kim, 2022), and materials discovery (Xie et al., 2022; Zeni et al., 2023). Our contributions are orthogonal to this literature as CONFHIT is model agnostic and offers guarantees for building property-satisfying samples from these generative models.

To summarize, compared with existing works, CONFHIT is the first framework that (i) eliminates the need for oracle access, (ii) corrects for covariate shift between historical and generated samples, and (iii) provides finite-sample guarantees for both certification and compact design in generative design.

## 2 PROBLEM FORMULATION

For expositional convenience, we slightly override the notations in Section 1.

**Data and distributions.** We assume access to a set of i.i.d. labeled (calibration) samples $\mathcal{D}_{\text{calib}} = \{(X_i, Y_i)\}_{i=1}^n$ from an unknown distribution $P$, e.g., molecules from past campaigns with known properties. Here $X_i \in \mathcal{X}$ is the feature, e.g., the chemical structure of a molecule, and $Y_i \in \{0, 1\}$ is a binary label, e.g., oracle-judged $Y_i = A(X_i)$. A generative model $\mathcal{M}$ produces i.i.d. samples $\{X_{n+j}\}_{j \geq 1}$ from another distribution $Q$ with unknown labels $\{Y_{n+j}\}_{j \geq 1}$ (the properties $Y_{n+j} = A(X_{n+j})$ should they be judged by the oracle). Although $Q$'s density is, in principle, computable, doing so is often costly. Instead, we assume a covariate shift between the two distributions: $\mathrm{d}Q/\mathrm{d}P(x, y) = w(x)$ for some function $w \colon \mathcal{X} \to \mathbb{R}^+$. It posits that $X$ captures all the information for predicting $Y$, which is especially sensible and standard in drug discovery contexts where the biological property is determined by the structure (though in a highly complex fashion).

**Guarantees.** We now formalize our guarantees for (i) certifying the presence of a hit in a set of generated candidates, and (ii) constructing a set of samples with at least a hit. For goal (i), given a pre-specified confidence level $\alpha \in (0, 1)$ and a generated set $\mathcal{C}_0^{\text{new}} := \{X_{n+j}\}_{j=1}^N$ of budget $N \in \mathbb{N}^+$, we aim to propose a test $\psi(\mathcal{D}_{\text{calib}}, \mathcal{C}_0^{\text{new}}) \in \{0, 1\}$ such that

$$\text{(Certification)} \qquad \mathbb{P}\big(\psi(\mathcal{D}_{\text{calib}}, \mathcal{C}_0^{\text{new}}) = 1 \,\big|\, A(X_{n+j}) = 0, \ \forall 1 \leq j \leq N\big) \leq \alpha, \qquad (1)$$

that is, the probability of falsely certifying a low-quality candidate set is upper bounded by $\alpha$. For goal (ii), we would like to find a (random) stopping point $\hat{N} \leq N$ such that

$$\text{(Design)} \qquad \mathbb{P}\big(A(X_{n+j}) = 0, \ \forall 1 \leq j \leq \hat{N}\big) \leq \alpha, \qquad (2)$$

outputing the list of samples $\hat{\mathcal{C}}^{\text{new}} := (X_{n+1}, \ldots, X_{n+\hat{N}})$ which contains a hit while controlling the error of vacuous declaration. We define $\hat{N} = 0$ to accommodate cases where we are not confident enough to declare any positive sample within the generation budget (failure of certification).[1]

## 3 METHOD

We introduce our conformal nested testing framework to address the certification problem (Section 3.1) and the design problem (Section 3.2), and prove their model-agnostic, finite-sample guarantees.

---

[1]This is consistent with existing works on uncertainty quantification for generative models (Quach et al., 2023), where a "null" option (i.e., the method fails to confidently generate good samples) is required.

### 3.1 CERTIFICATION: JOINT WEIGHTED CONFORMAL P-VALUE

We begin by addressing the certification problem: *Given a set of generated samples, how to quantify the confidence in it containing a hit?* Our key strategy is to construct a p-value based on conformal inference ideas that test the presence of any hit in $\{X_{n+j}\}_{j=1}^N$. Recall that there is a covariate shift $w(x) = \mathrm{d}Q/\mathrm{d}P(x,y)$ for the labeled calibration data and the new test samples. To fix ideas, we assume the density ratio $w(\cdot)$ is known for now, and discuss estimated density ratio in Remark 3.3.

Our p-values leverage the inactive calibration data $\{X_i \colon i \in \mathcal{I}_0\}$ where $\mathcal{I}_0 := \{i \in [n] \colon Y_i = 0\}$. We let $\boldsymbol{X} = (X_1, \ldots, X_{n_0}, X_{n+1}, \ldots, X_{n+N})$ denote the vector of (inactive) calibration and test covariates. Following split conformal prediction (Vovk et al., 2005), we let $V \colon \mathcal{X}^{n_0+N} \to \mathbb{R}$ be any function whose training process is independent of $\{(X_i, Y_i)\}_{i \geq 1}$, which is then viewed as fixed. We call $V$ the *conformity score* function, whose choice is discussed near the end of this subsection. Without loss of generality, we assume a larger value of the conformity score indicates stronger confidence in the presence of a positive value in $\{Y_{n+j}\}_{j=1}^N$.

To address multiple test points without introducing too much computational overhead, we leverage randomization (see Appendix A.1 for the non-randomized version). Let $\Pi_N$ be the collection of all permutations of $\{1, \ldots, n_0, n+1, \ldots, n+N\}$, and denote the identity mapping as $\pi^{(0)}$, i.e., $\pi^{(0)}(i) = i$ for any $i$. Formally, fixing any $B \in \mathbb{N}^+$, we draw $\pi^{(1)}, \ldots, \pi^{(B)} \overset{\text{i.i.d.}}{\sim} \mathrm{Unif}(\Pi_N)$, the uniform distribution over the permutation space. Define

$$p_N^{\mathrm{rand}} = \frac{\sum_{b=0}^B \bar{w}(\pi^{(b)}; \boldsymbol{X}) \mathbb{1}\{V(\pi_0; \boldsymbol{X}) \leq V(\pi^{(b)}; \boldsymbol{X})\}}{\sum_{b=0}^B \bar{w}(\pi^{(b)}; \boldsymbol{X})}. \tag{3}$$

Here, for any permutation $\pi \in \Pi_N$, we define the conformity score from the permuted data $V(\pi; \boldsymbol{X}) = V(X_{\pi(1)}, \ldots, X_{\pi(n_0)}, X_{\pi(n+1)}, \ldots, X_{\pi(n+k)})$, and $\bar{w}(\pi; \boldsymbol{X}) = \prod_{j=1}^k w(X_{\pi(n+j)})$, is the joint likelihood ratio after permutation $\pi$, where $w(x) = \mathrm{d}Q/\mathrm{d}P(x,y)$ is the density ratio.

The construction of our p-value relies on an analysis of the exchangeability structure between the inactive calibration data $\{X_i \colon i \in \mathcal{I}_0\}$ and the test samples $\{X_{n+j}\}_{j=1}^N$ under distribution shift, detailed in Appendix A.2 due to limited space. It extends the conformal p-values in the literature via an extended weighted exchangeability structure (Tibshirani et al., 2019) for multiple test samples (Hu & Lei, 2024; Jin & Candès, 2023a); these connections are discussed in Appendix A.3.

The following theorem establishes the finite-sample validity of the randomization p-value (3). Notably, this holds regardless of the number of sampled permutations. Its proof is in Appendix B.4.

**Theorem 3.1.** *Under the covariate shift assumption, it holds for any fixed $t \in [0,1]$ that*

$$\mathbb{P}\big(p_N^{\mathrm{rand}} \leq t \,\big|\, \max_{1 \leq j \leq N} Y_{n+j} = 0\big) \leq t.$$

*Therefore, the certification test function $\psi(\mathcal{D}_{calib}, \{X_{n+j}\}_{j=1}^N) = \mathbb{1}\{p_N^{\mathrm{rand}} \leq \alpha\}$ achieves* (1).

Inheriting the model-free nature of conformal inference, the validity of $p_k$ and $p_k^{\mathrm{rand}}$ holds regardless of the score function $V$, as long as its training process is independent of the calibration and test data.

**Remark 3.2** (Outlier detection). *Our p-values can also be viewed as testing for the presence of at least one outlier in $\{X_{n+j}\}_{j=1}^m$ when $\{X_i\}_{i \in \mathcal{I}_0}$ are viewed as calibration inliers. We refer the readers to discussion on the connection to outlier detection under covariate shifts (Bates et al., 2023; Jin & Candès, 2023a) in Appendix A.4.*

**Remark 3.3** (Estimated density ratio). *In many practical scenarios, the density ratio $w(\cdot)$ is unknown or difficult to evaluate. In such cases, we estimate the density ratio by $\hat{w}(\cdot)$, and use $\hat{w}(\cdot)$ instead of $w(\cdot)$ in the construction of our p-values $\{p_k\}$. For instance, we can train the density ratio over a random subset of all the labeled data and independently generated test samples. We do not write the exact formulas for such plug-in p-values for brevity. Recognizing that the success of CONFHIT hinges on accurate density ratio estimation, we provide a thorough discussion on the robustness of CONFHIT with estimated weights and several practical diagostics in Section 3.3.*

**Conformity score function.** Because we certify the candidate set when $p_N$ is small, $V$ must grow when $X_{n+1:n+N}$ contains a positive to indicate strong confidence. To this end, we assume access to a pre-trained model $\hat{\mu} \colon \mathcal{X} \to \mathbb{R}$ that predicts the unknown property $Y_{n+j}$ based on $X_{n+j}$, which will be used in the score $V$. We suggest several natural choices: (i) Max-pooling: $V(x_{1:n_0}, x_{n+1:n+N}) =$

$\max_{1 \le j \le N} \hat{\mu}(x_{n+j})$, (ii) Sum-of-prediction: $V(x_{1:n_0}, x_{n+1:n+N}) = \sum_{j=1}^{N} \hat{\mu}(x_{n+j})$, (iii) Rank-sum: $V(x_{1:n_0}, x_{n+1:n+N}) = \sum_{j=1}^{N} R_{n+j}$, where $R_{n+j}$ is the rank among all scores, (iv) Likelihood ratio: $V(x_{1:n_0}, x_{n+1:n+N}) = \sum_{j=1}^{N} \log(\frac{\hat{\mu}(x_{n+j})}{1-\hat{\mu}(x_{n+j})})$. We explain these scores in Appendix A.5. We note that CONFHIT is model-agnostic: practitioners can choose any suitable model to build the score $V$. The quality of the score/model affects the power, but not the error control of CONFHIT. Finally, practitioners may want to avoid sample splitting when labeled data is limited; we provide such a variant in Appendix A.6 where the labeled data are used for both training $\hat{\mu}$ and p-value construction.

## 3.2 DESIGN: CONFORMAL NESTED TESTING

Upon the valid p-value and certification test for a given candidate set, we proceed to address the design problem: *How to propose a compact set of generated samples that contains at least one hit with high probability?* To this end, we connect (2) to the problem of constructing a smaller subset of candidates that can be certified by our procedure, and establish its statistical guarantee.

To be specific, for every $1 \le k \le N$, we consider the null hypothesis

$$H_k : Y_{n+j} = 0, \ \forall \ 1 \le j \le k. \tag{4}$$

That is, $H_k$ posits that none of the first $k$ generated instances obeys the desired property. From a hypothesis testing perspective, rejecting $H_k$ thus suggests sufficient confidence in declaring a hit in $\{X_{n+j}\}_{j=1}^{k}$. The certification strategy in Section 3.1 is readily applicable to obtain a p-value $p_k$ for certifying each subset $\{X_{n+j}\}_{j=1}^{k}$ indexed by $k$ that obeys $\mathbb{P}(p_k \le t \,|\, H_k \text{ is true}) \le t$ for $t \in [0, 1]$.

Our solution to the design problem appears simple: determine an index $\hat{N}$ such that $H_{\hat{N}}$ can be rejected (i.e., $\{X_{n+j}\}_{j=1}^{k}$ can be confidently declared as containing a hit). Specifically, suppose we can construct a *decreasing* sequence of p-values $p_k \in [0, 1]$ for each fixed $H_k$. Then, we set

$$\hat{N} := \inf\{k : p_k \le \alpha\}. \tag{5}$$

That is, we take the first p-value that passes the significance level $\alpha \in (0, 1)$. Theorem 3.4 confirms the validity of this nested testing strategy, whose proof is in Appendix B.1.

**Theorem 3.4.** *Suppose the p-values $p_k \in [0, 1]$ obey: (i) Monotonicity: $p_1 \ge p_2 \ge \cdots p_N$. (ii) Validity: For any fixed $k \ge 1$ and $t \in [0, 1]$, it holds that $\mathbb{P}(p_k \le t \,|\, H_k \text{ is true}) \le t$. Then, we have*

$$\mathbb{P}(\max_{1 \le j \le \hat{N}} Y_{n+j} = 0) \le \alpha$$

*for the generation threshold $\hat{N}$ computed as in (5). Here $H_k$ is defined in (4).*

We note that any p-values $\{\tilde{p}_k\}$ obeying condition (ii) in Theorem 3.4, such as those constructed in the certification problem, can be turned to $p_k = \max_{k \le j \le N} \tilde{p}_j$ which satisfy both conditions in Theorem 3.4. Perhaps surprisingly, even though we are simultaneously examining multiple batches of candidates, it turns out that using a *monotone* sequence of *individually* valid p-values suffices to achieve our goal, and no adjustment for multiplicity is needed. We remark that the key to our theoretical guarantee is the nested nature of the hypotheses: if $H_k$ is true, then all "earlier" hypotheses $\{H_\ell\}_{\ell \le k}$ must also be true. We thus call our method "conformal nested testing".

**CONFHIT: Putting everything together.** So far, we have completed all the elements of CONFHIT. We summarize the entire procedure in Algorithm 1, including both certification (for every nested subset) and design. Note that the input $\hat{w}(\cdot)$ denotes an estimated density ratio function, which coincides with $w(\cdot)$ when the density ratio is known. When it fails to certify even the largest batch, CONFHIT flags "not confident enough" to clearly communicate the difficulty of the generation task.

## 3.3 ROBUSTNESS AND DIAGNOSTICS FOR DENSITY RATIO ESTIMATION

The density ratio often needs to be estimated in practice and its quality naturally affects the performance of CONFHIT. The difference between historical assay data and newly generated molecules is a persistent issue in drug discovery and needs to be addressed in nearly all conformal prediction methods in this domain (Krstajic, 2021; Laghuvarapu et al., 2023; Jin & Candès, 2023a; Fannjiang & Park, 2025). Estimating the density ratio is a statistically standard problem used in many fields including machine learning, statistics, and causal inference with rich existing toolboxes such as kernel or classification-based methods beyond the kernel density estimation in our experiments (Horvitz &

---

**Algorithm 1** CONFHIT

---

**Input:** Calibration data $\{X_i\}_{i=1}^{n_0}$ for which $Y_i = 0$, test data $\{X_{n+j}\}_{j=1}^{N}$, conformity score $V: \mathcal{X}^{n_0+k} \to \mathbb{R}$, (estimated) weight function $\hat{w}: \mathcal{X} \to \mathbb{R}^+$, confidence level $\alpha \in (0,1)$, budget $N \in \mathbb{N}^+$, Monte Carlo size $B \in \mathbb{N}^+$.

1: **for** $k = 1, \ldots, N$ **do**
2:      Set $\Pi = \{$ all permutations of $\{1, \ldots, n_0, n+1, \ldots, n+k\} \}$.          // Sequential conformal p-values
3:      Draw $\pi^{(b)} \overset{\text{i.i.d.}}{\sim} \text{Unif}(\Pi)$ for $b = 1, \ldots, B$.
4:      Compute p-value $p_k$ as in (3) with $w(x) := \hat{w}(x)$.
5: **end for**
6: Set $p_k = \max_{k' \geq k} p_{k'}$ iteratively for $k = N-1, \ldots, 1$          // Monotonize p-values
7: Compute $\hat{N} = \text{argmax}\{k \in [N]: p_k \leq \alpha\}$, with $\hat{N} = 0$ if $p_N > \alpha$

**Output:** Confident shortlist $\mathcal{C} := \{X_{n+j}\}_{j=1}^{\hat{N}}$ or "not confident enough" when $\hat{N} = 0$.

---

Thompson, 1952; Shimodaira, 2000; Sugiyama et al., 2007; Nguyen et al., 2010; Sugiyama et al., 2012; Shapiro, 2003). Below, we elaborate on (i) a theoretical analysis of CONFHIT's robustness to the estimation error, followed by (ii) three diagnostic approaches for the estimation quality.

Theorem 3.5 shows that with estimated density ratio, the error inflation depends on how *aggressive* (resp. *conservative*) the weights are for small (resp. *large*) p-values; its proof is in Appendix B.5.

**Theorem 3.5** (Robustness to estimation error). *Let $\hat{p}_N$ be the p-value (3) using any estimated weights $\hat{w}(\cdot)$. Define the positive/negative parts of the estimation error $\hat{\delta}^{\pm}(\pi; \boldsymbol{X}) = [\hat{\bar{w}}(\pi; \boldsymbol{X}) - \bar{w}(\pi; \boldsymbol{X})]_{\pm}$, with $\hat{\bar{w}}(\pi; \boldsymbol{X}) = \prod_{j=1}^{N} \hat{w}(X_{\pi(n+j)})$. Then, for any fixed $t \in (0,1)$, we have*

$$\mathbb{P}\left(\hat{p}_N^{rand} \leq t \,\Big|\, \max_{1 \leq j \leq N} Y_{n+j} = 0\right) \leq t + \mathbb{E}\left[\frac{\hat{t} \cdot \hat{\Delta}^+ + (1-\hat{t}) \cdot \hat{\Delta}^-}{\sum_{b=0}^{B} \bar{w}(\pi^{(b)}; \boldsymbol{X})} \,\Big|\, \max_{1 \leq j \leq N} Y_{n+j} = 0\right],$$

*where, setting $\hat{p}_N(v) := \frac{\sum_{b=0}^{B} \hat{\bar{w}}(\pi^{(b)}; \boldsymbol{X}) \mathbb{1}\{v \leq V(\pi^{(b)}; \boldsymbol{X})\}}{\sum_{b=0}^{B} \hat{\bar{w}}(\pi^{(b)}; \boldsymbol{X})\}}$, we define $\hat{v} = \text{argmax}\{V(\pi^{(b)}; \boldsymbol{X}): \pi \in \Pi, \hat{p}_N(V(\pi^{(b)}; \boldsymbol{X})) \leq t\}$ as the score cutoff, and the corresponding p-value as $\hat{t} := \hat{p}_N(\hat{v})$, and*

$$\hat{\Delta}^+ = \sum_{b=0}^{B} \hat{\delta}^+(\pi^{(b)}; \boldsymbol{X}) \mathbb{1}\{V(\pi^{(b)}; \boldsymbol{X}) < \hat{v}\}, \quad \hat{\Delta}^- = \sum_{b=0}^{B} \hat{\delta}^-(\pi^{(b)}; \boldsymbol{X}) \mathbb{1}\{V(\pi^{(b)}; \boldsymbol{X}) \geq \hat{v}\}.$$

While Theorem 3.5 offers insights on how the guarantees degrade with the (unknown) estimation error, it is often useful to perform robustness diagnostics in practice. The three approaches below are demonstrated in our experiments in "Robustness check" Section 4.5.

**(1) Balance check.** Following the balancing idea in causal inference (Ben-Michael et al., 2021), reweighting via the correct density ratio will bring the empirical mean of any function of the covariates in the calibration data close to that of the test data (under the null with no hits). A natural diagnostic is to check the imbalance $|\frac{1}{n_0} \sum_{i=1}^{n_0} \hat{w}(X_i) f(X_i) - \frac{1}{k} \sum_{j=1}^{k} f(X_{n+j})|$ for some $f: \mathcal{X} \to \mathbb{R}$.

**(2) Validation shift.** A second approach is to examine the reliability of estimation via synthetic shift. One may create artificial distribution shifts in the labeled data, such as by scaffold splitting; with one fold as the calibration data and the other as the test data, one can use the known labels to examine the uniformity of p-values and the quality of the density ratio estimation algorithm.

**(3) Sensitivity analysis.** Finally, one may conduct sensitivity analysis to probe the robustness of CONFHIT to perturbations of estimated weights. Similar to causal inference (Rosenbaum, 2005), we may assume the correct weights are within a distance from the estimated weights, and examine the worst-case p-values and testing results while varying the weights within the assumed range.

While generative models do not naturally provide a built-in notion of reliability, CONFHIT augments them with new capabilities for practical deployment; it is model-agnostic and only relies on a scalar quantity, the density ratio, that can be validated and stress-tested, thereby making the uncertainty measurable and more robust than trusting the raw model alone (Section 4.5, "Additional baselines").

## 4 EXPERIMENTS

### 4.1 TASKS, MODELS, AND DATASETS

We evaluate our framework on two representative conditional molecule-generation tasks and highlight the model-agnostic nature of our approach by employing five generative models spanning VAE,

autoregressive transformers, diffusion models, and Bayesian Flow Networks. For *evaluation* purposes, we use computational oracles to judge the property of generated samples since wet-lab evaluation is infeasible (Jin et al., 2020); the oracle is not used in running CONFHIT.

**App. 1: Constrained Molecule Optimisation (CMO).** Given a seed molecule $\bar{X}$ that *fails* a target property, the goal is to generate a molecule $X$ that is close in Tanimoto distance to $\bar{X}$ and satisfies the property. This task reflects lead-optimisation workflows where *experimental* assays provide ground-truth. Following the standard practice of using a computational oracle, in our experiments, we use the property *DRD2 receptor binding (DRD2)*, with success ($Y = 1$) if $\mathrm{DRD2}(x) > 0.5$. Additional comphrehensive results on Quantitative Estimate of Drug-likeness improvement (QED) reported in the Appendix E.2.

To demonstrate the compatibility of CONFHIT with general generative models, we employ two state-of-the-art CMO models: 1. *H*GRAPH2GRAPH, a hierarchical VAE editing scaffold-level motifs while preserving validity (Jin et al., 2020); 2. *SELF*-EDIT, an autoregressive transformer generating SELFIES strings to optimise the property while maintaining similarity to the seed (Jiao et al., 2023).

We take a subset of the training split in Chemprop (Yang et al., 2019) of ChEMBL data (Gaulton et al., 2012) as the labeled calibration data $\{\bar{X}_i, X_i, Y_i\}_{i=1}^n$ (seed input, sample from past campaigns, and oracle property), following Jin et al. (2020). Inactive molecules in the test split serve as test inputs $\{\bar{X}^{(\ell)}\}_{\ell=1}^L$. For each $\bar{X}^{(\ell)}$, the model generates $\{X_{n+j}^{(\ell)}\}_{j=1}^m$.

**App. 2: Structure-Based Drug Discovery (SBDD).** Here the input $\bar{X}$ is a 3D protein binding pocket, and the task is to generate ligands $X$ that bind to it, so the label $Y$ depends on $(\bar{X}, X)$. Following standard practice, for evaluation purposes only, we use AutoDock Vina (Trott & Olson, 2010) as computation oracle and label a ligand as a hit if its score is below $-7.5 \, \mathrm{kcal \, mol^{-1}}$; such a threshold captures about $75\%$ of known active samples in CrossDock (Francoeur et al., 2020a).

In this task, we apply CONFHIT with three state-of-the-art SBDD models: *TargetDiff* (Guan et al., 2023), an SE(3)-equivariant diffusion model conditioned on pocket meshes; *DecompDiff* (Guan et al., 2024), using decomposed priors to separate structural components; *MolCRAFT* (Qu et al., 2024), a Bayesian Field Network operating fully in continuous parameter space.

The calibration data are protein-ligand pairs $\{(\bar{X}_i, X_i)\}_{i=1}^n$ from CrossDocked (Francoeur et al., 2020b). Following the same split in Guan et al. (2023), we use the proteins in the test split as inputs $\{\bar{X}^{(\ell)}\}_{\ell=1}^L$, and the model generates candidate ligands $\{X_{n+j}\}_{j=1}^N$ given each input $\bar{X}^{(\ell)}$.

## 4.2 IMPLEMENTATION DETAILS

**Conformity Score Function.** We define $V$ using a property prediction model $\hat{\mu}(\cdot)$:

- **CMO.** For both properties (QED and DRD2) we train a binary classifier $\hat{\mu}(\cdot)$ with molecular graph inputs using the Chemprop library (Yang et al., 2019) and ChEMBL data splits in Jin et al. (2020).
- **SBDD.** We train an EGNN $\hat{\mu}(\cdot)$ (Satorras et al., 2021) on PDBBind crystal structure data (Wang et al., 2005) to predict binding affinity, using TARGETDIFF (Guan et al., 2023) codebase.

In both tasks, $\hat{\mu}$ is trained on data independent of calibration and test sets. Training details, hyperparameters, and compute for $\hat{\mu}$ and the generative models are given in Appendix C. For conciseness, we present most results with the Max-pooling score $V(x_{1:n_0}, x_{n+1:n+k}) = \max_{1 \le j \le k} \hat{\mu}(x_{n+j})$. Comparisons to other score functions appear in Figure 4.

**Density Ratio Estimation.** We leverage kernel density estimation (KDE) for constructing the weights in our p-values. Following the discussion in Lemma A.2 in Appendix A.2, the desired weight $w(\cdot)$ equals the *marginal* density ratio of generated samples, regardless of hit status. Also, plugging in the densities for the joint distribution of input/generation pairs, denoted as $\tilde{q}(\bar{x}, x)$ and $\tilde{p}(\bar{x}, x)$, yields the same p-value due to normalization. Thus, we directly estimate $\tilde{p}(\cdot)$ and $\tilde{q}(\cdot)$ to ensure sufficient data for estimation. Following CODRUG (Laghuvarapu et al., 2023), we extract latent features $g(\bar{x}, x)$ from the penultimate layer of the property model, and fit Gaussian KDEs $\hat{p}(\bar{x}, x)$ and $\hat{q}(g(\bar{x}, x))$ on calibration and test samples, forming weights $\tilde{w}(\bar{x}, x) = \hat{q}(g(\bar{x}, x))/\hat{p}(g(\bar{x}, x))$.

## 4.3 RESULTS: CERTIFICATION

In this section, we present results of the certification experiment, where the goal (1) is to certify the presence of a hit in a batch of fixed size $N$ at level $\alpha$. Our evaluation focuses on (i) error control, and (ii) power, i.e., the frequency of successfully certifying hits. Results for both CMO and SBDD tasks

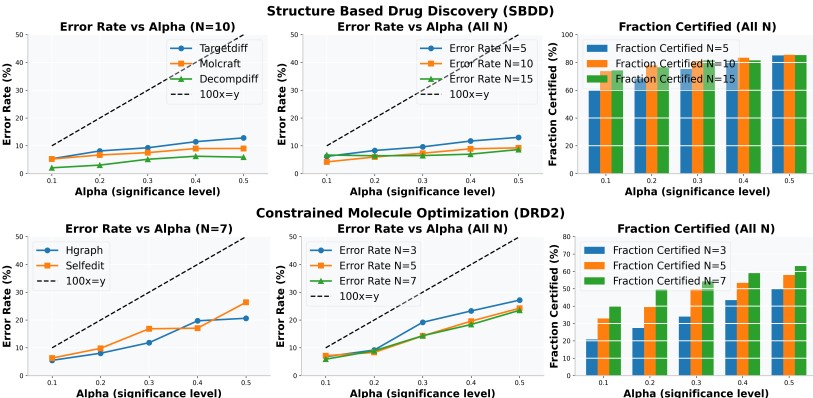

Figure 2: **Certification results.** Left: realized error rates at fixed $N$ for different models and error levels $\alpha$ in SBDD (upper) and CMO (lower). Middle: average error rates while varying budget $N$. Right: power, i.e., the fraction of actives certified at various error level $\alpha$ and budget $N$ values. The dashed line denotes the ideal $y = x$ error bound. Our method consistently achieves valid coverage across scenarios. Results are averaged over 5 random runs; error bars and additional results for other values of $N$ are in Appendix E.2

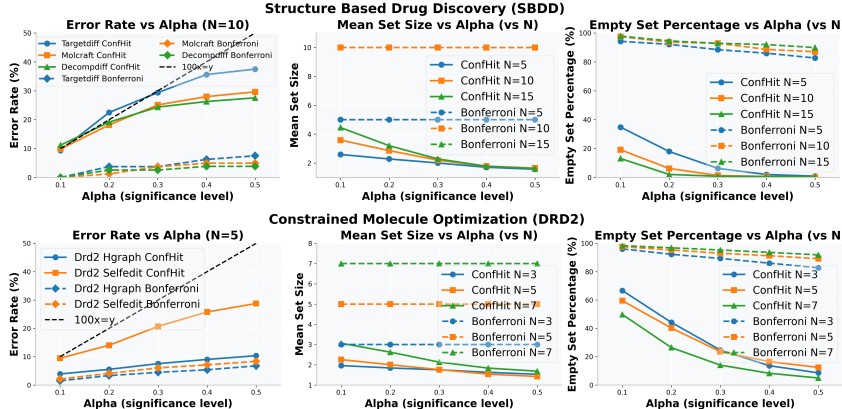

Figure 3: **Design results.** Error rate at fixed $N$ for different methods (left), mean set sizes averaged across methods at different values of $N$ (middle), and empty set percentage at different values of $N$ (right) across target levels $\alpha$. The top row shows results for SBDD and the bottom for CMO. Dashed black line in the error plots indicates the ideal $y = x$ bound. CONFHIT achieves tight error control while producing substantially smaller sets. Results are averaged over 5 random runs; additional results are provided in the Appendix E.2.

across different generative models are shown in Figure 2. The left panel reports the empirical error rate, i.e. the frequency of certifying sets that contain no hit. Across all settings, we observe tight error control. The middle panel shows the error rates (averaged across methods for visualization) for different budgets $N$, and we observe that the error stays below the target error levels. Finally, the right panel plots the fraction of sets containing an active that are certified as a function of $\alpha$ at different, where CONFHIT is able to detect true hits with satisfactory power. As expected, this fraction increases with the significance level and set size $N$.

## 4.4 RESULTS: DESIGN

In this section, we present experiments for the design problem in (2), where we prune a generated batch to obtain compact subsets while preserving tight error control. As summarized in Figure 3, our nested procedure refines candidate sets while preserving statistical guarantees.

As we address a novel problem under unique constraints, there are no directly comparable baselines (existing methods in conformal generative modeling (Quach et al., 2023; Kladny et al., 2024) are not comparable due to the unavailable oracle). Instead, we consider two baselines to demonstrate the benefits of CONFHIT: **(i) Bonferroni correction.** Given $N$ test samples, we threshold the one-test-sample p-value (set each candidate as the test set in (3)) at $\alpha/N$. It provides a stronger guarantee of no false certification for any test sample, but can be conservative. **(ii) Certification-only.** The certification procedure from Section 4.3, which serves as a baseline without pruning.

**Compared with Bonferroni,** our approach substantially outperforms in both CMO and SBDD tasks. At stringent levels ($\alpha = 0.1$), Bonferroni yields empty sets for nearly 100% of SBDD inputs, while CONFHITleads to empty set frequency as low as 16% for $N = 15^2$. Moreover, in the rare cases where Bonferroni is able to make discoveries, it usually exhaust the full budget, whereas CONFHIT consistently prunes to 2-5 molecules (30-50% of the full budget). Due to conservativeness, Bonferroni has low or near zero error rates, whereas CONFHIT achieves adaptive and tight coverage.

**Compared with certification-only** whose results are in Figure 2, the nested procedure successfully prunes promising initial large sets to smaller ones. For example, in SBDD at $\alpha = 0.1$ with $N = 15$, the nested procedure reduces the mean set size to about 4 molecules without compromising the error control. Similar improvements hold in CMO, where nested pruning typically halves the certified set size while delivering tighter agreement between realized and nominal error.

In summary, CONFHIT delivers compact certified sets with strict error control. It largely avoids vacuous results and yields smaller, more actionable shortlists with higher coverage for subsequent experimental validation. This makes our framework particularly valuable in scientific discovery workflows, where experimental budgets require both statistical confidence and practical tractability.

### 4.5 ADDITIONAL INVESTIGATIONS

**Choice of test statistic.** While we report the results with the Max-pool conformity score, Figure 4 evaluate CONFHIT with five choices of the conformity score $V(\cdot)$—*min*, *mean*, likelihood-ratio (*LR*), rank-sum, and *max*—on the SBDD task with TARGETDIFF at a budget of $N = 5$; results for other tasks and budgets show similar patterns and are provided in the Appendix E.2. Across the board, CONFHIT consistently keeps the realised error rate close to the target $\alpha$, confirming the model-agnostic validity. However, the choice of the score does affect power: *max* certifies most often (e.g., 73 % of seeds at $\alpha = 0.1$), while rank-sum is the most conservative.

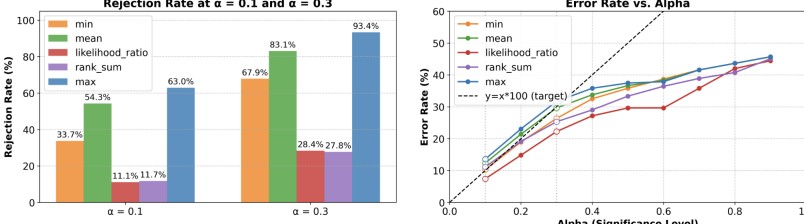

Figure 4: Comparison of score statistics on TargetDiff (SBDD, N=5). Left: rejection (power) at $\alpha = 0.1, 0.3$. Right: error vs. $\alpha$. All remain valid; the *max* statistic shows the highest power.

**Choice of property prediction model.** As we discussed in the end of Section 3.1, the choice of the property prediction model only affects the power. We performed an ablation study with weak property prediction models by adding noninformative perturbations to the predictions. CONFHIT maintains robust error rate control despite that the weaker predictor decreases the power; see Appendix F.

**Distribution shift adjustment.** The guarantees of CONFHIT in both certification and design rely on valid p-values, which further hinges on distribution shift adjustment. As an ablation study, we compare CONFHIT with the version without distribution shift adjustment, i.e., taking $w(\cdot) \equiv 1$. We observe that running CONFHIT without distribution shift adjustment can violate the coverage guarantee especially for stringent target error rates, for example, using the TARGETDIFF model in the SBDD task, and the SELFEDIT model in the CMO task. The differences are highlighted in Figure 5a. In addition, we plot the distribution of p-values computed with negative samples across different datasets and observe that they are approximately uniform; see Appendix E.3.

**Robustness check.** We exemplify the robustness diagnostics in Section 3.3, with a sensitivity-analysis in Appendix H.1, synthetic validation with a realistic scaffold split in Appendix H.2 and a balance check based on key features in Appendix H.3. First, by perturbing the weights via a power law, we showcase how to examine the robustness of CONFHIT's output, with additional evaluation to show the mild degradation of error control. Meanwhile, CONFHIT achieves tight error control under synthetic scaffold splits, and the estimated density ratio improves the balance in key features.

---

$^2$Returning empty sets means no enough confidence under the generation budget $N$, which is inevitable without strong assumptions on the generative power.

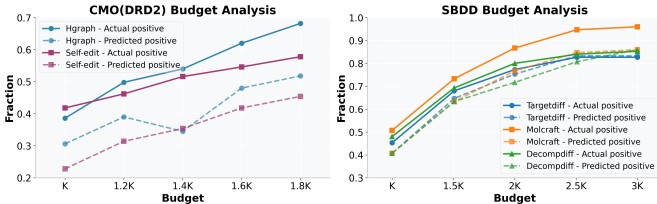

Figure 5: (a) **Distribution shift adjustment** (left). Coverage violations when CONFHIT is run without density correction. (b) **Budget analysis** (middle and right). Fraction of inputs with at least one hit under increasing generation budget. Solid lines: fraction of actual hits; dashed lines: predicted fraction of hits.

**Additional baselines.** We additionally evaluate two baselines. (i) A **heuristic baseline** which uses an estimated probability to calibrate the generation batch size (Appendix G.1). It violates error control when the estimation quality moderately degrades; in contrast, CONFHIT is model-agnostic and maintains validity under the same degradation (Appendix F), showing that principled uncertainty quantification provides robust guarantees. (ii) An **oracle baseline** which (unrealistically) labels generated samples (Quach et al., 2023) yet without covariate shift adjustment (Appendix G.2). Even with oracle access, ignoring the distribution shift leads to both high errors and high frequency of empty sets (perhaps since it relies on exchangeability between *inputs*, which reduces the sample size).

## 4.6 WORKING UNDER AN OVERALL BUDGET

Finally, we demonstrate the utility of CONFHIT for allocating a fixed budget across multiple tasks. In drug discovery campaigns, scientists often work with multiple generation inputs but can only validate a limited number of samples experimentally. In this setting, the confidence in valid hits provided by CONFHIT serves as a practical heuristic for budget allocation—deciding how many candidates to synthesize per task. Given $K$ inputs (tasks) and a total budget $B$, we fix a maximum size $N$ for each input; then, CONFHIT suggests the smallest subset which contains a hit for each input at a confidence level $\alpha$. We then vary the value of $\alpha$ until the budget is exhausted. Finally, we estimate the fraction of tasks whose output set contains a hit via $(1 - \alpha)$ minus the fraction of empty sets (i.e., when CONFHIT fails to certify a hit at level $\alpha$). The exact procedure is described in Appendix D.

In Figure 5b, we compare predicted and realized positives under different budget levels (measured as multiples of the number $K$ of inputs). Across both CMO and SBDD, the realized fraction of actives consistently exceeds our estimates, showing the reliability and practical utility of CONFHIT. Complementing the certification and design results, this budge allocation perspective confirms the robust empirical performance of CONFHIT when coupled with more complex, albeit heuristic, applications. Of course, we acknowledge that this is only a heuristic approach without formal theoretical guarantees, and the application of CONFHIT in such tasks warrants careful future study.

## 5 CONCLUSION

In this work, we introduced CONFHIT, a model-agnostic framework that delivers finite-sample guarantees for conditional generative models. By re-weighting calibration data with density ratios, CONFHIT corrects the distributional shift from historical compounds to model-generated candidates, thereby removing the need for an external experimental oracle, an assumption required by all existing conformal-prediction approaches in generative modeling. Building on these adjusted weights, we derive a nested conformal p-value that certifies the probability of sampling at least one viable molecule. Across two standard benchmarks for optimization of constrained molecules and structure-based design, and across a wide range of methods and budget regimes, CONFHIT consistently provides valid coverage guarantees while maintaining compact certified sets. Our results establish CONFHIT as a principled and practical framework for both certification and design, enabling reliable generative modeling under stringent resource constraints.

**Limitations** The coverage guarantee relies on density-ratio estimates, which can be noisy when the calibration set is small or the feature extractor is poorly aligned with the target domain. Our experiments focus on small molecules; extending the approach to proteins or other macromolecules with larger, more structured generative space will require additional work. Validation currently relies on in-silico oracles; without wet-lab confirmation, transferability remains to be seen. Demonstrating robustness across broader chemical and experimental settings is a key direction for future research.

## 6 ETHICS STATEMENT

Because the framework is model-agnostic and targets conditional molecule generation, we expect it to benefit diverse discovery pipelines in chemistry, biology, and materials science. The work builds on long-standing practices in molecular modeling, so we do not anticipate negative societal consequences. However, we acknowledge potential dual use, such as designing harmful compounds, and responsible use will require oversight and access controls.

**LLM Usage** The use of LLMs in this work is limited to polishing the writing and assisting with code-related tasks.

## 7 REPRODUCIBILITY STATEMENT

All experimental settings are described in Section 4.1, with an extended discussion of implementation details in Appendix C. Information on compute resources and runtime is provided in Section C.4. The code, datasets along with the config files used for experiments in this work are included at https://github.com/siddharthal/CONFHIT-Conformal-Generative-Design-with-Oracle-Free-Guarantees and are further detailed in Appendix C.5.

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

# A  ADDITIONAL DISCUSSION

## A.1  NON-RANDOMIZED CONFORMAL P-VALUE

Let $\Pi$ be the collection of all permutations of $\{1, \ldots, n_0, n+1, \ldots, n+k\}$. For any permutation $\pi \in \Pi$, we denote the score obtained by permuting the observed features by $\pi$ as
$$V(\pi; \boldsymbol{X}) = V(X_{\pi(1)}, \ldots, X_{\pi(n_0)}, X_{\pi(n+1)}, \ldots, X_{\pi(n+k)}).$$

We additionally denote the identity mapping as $\pi_0$, i.e., $\pi_0(i) = i$ for any $i$. Finally, we define

$$p_k = \frac{\sum_{\pi \in \Pi} \bar{w}(\pi; \boldsymbol{X}) \, \mathbb{1}\{V(\pi_0; \boldsymbol{X}) \leq V(\pi; \boldsymbol{X})\}}{\sum_{\pi \in \Pi} \bar{w}(\pi; \boldsymbol{X})}. \tag{6}$$

Here, $\bar{w}(\pi; \boldsymbol{X}) = \prod_{j=1}^{k} w(X_{\pi(n+j)})$, the joint likelihood ratio after permutation $\pi$, where $w(x)$ is the density ratio. In a nutshell, $p_k$ considers all permutations of the inactive calibration samples and the test samples, and compute the weighted rank of the score computed with observed data, $V(\pi_0; \boldsymbol{X})$, among all scores with permuted data.

The validity of the deterministic p-value is established in the following theorem, whose proof is in Appendix B.3.

**Theorem A.1.** *Under the covariate shift assumption, it holds for any fixed $t \in [0, 1]$ that*

$$\mathbb{P}\big( p_k \leq t \,\big|\, max_{1 \leq j \leq k} Y_{n+j} = 0\big) \leq t.$$

## A.2 JOINT WEIGHTED EXCHANGEABILITY

In this part, we characterize the exchangeability structure of the inactive calibration data $\{X_i\}_{i \in \mathcal{I}_0}$ and the test samples $\{X_{n+j}\}_{j=1}^{k}$ under the null case

$$H_k: \max_{1 \le j \le k} Y_{n+j} = 0.$$

Here for generality, we use the index $k$ to indicate that $\{X_{n+j}\}_{j=1}^{k}$ might be a subset of the original proposal. The following lemma characterizes the distribution of these data conditional on the labels. Its proof simply applies the Bayes' rule, whose proof is deferred to Appendix B.2 for completeness.

**Lemma A.2.** *Conditional on $\{Z_i\}_{i=1}^{n} \cup \{Z_{n+j}\}_{j \ge 1}$ as well as $\max_{1 \le j \le k} Y_{n+j} = 0$, the features $\{X_i\}_{i \in \mathcal{I}_0}$ are i.i.d. from $P_0 := P_{X\,|\,Y=0}$, whereas $\{X_{n+j}\}_{j=1}^{k}$ are i.i.d. from $Q_0 := Q_{X\,|\,Y=0}$. Furthermore, $\mathrm{d}Q_0/\mathrm{d}P_0(x) = w(x)$, where $w(x) = \frac{\mathrm{d}Q}{\mathrm{d}P}(x,y) = \frac{\mathrm{d}Q_X}{\mathrm{d}P_X}(x)$ is the density ratio.*

Lemma A.2 states that, conditional on all labels, on the "null" event that $H_k$ is true, the "marginal" density ratio $w(x)$ suffices to adjust for distribution shifts between the inactive calibration data in $\mathcal{I}_0$ and the newly generated data. We discuss its implications on density ratio estimation below.

**Remark A.3.** *When $w(\cdot)$ is unknown, Lemma A.2 justifies using a random subset of the labeled data and an independent set of generated samples (no matter whether they are active) to estimate $w(\cdot)$. An alternative approach is to directly estimate the density ratio between $\{X_i\}_{i \in \mathcal{I}_0}$ and $\{X_{n+j}\}_{j=1}^{k}$. However, this approach leads to quite limited sample sizes when $k$ is small.*

Lemma A.2 allows to characterize the exchangeability structure among calibration and test points, extending the weighted exchangeability notion (Tibshirani et al., 2019) to multiple test points. It is a consequence of a conditional permutation-style result which extends foundational results in conformal prediction (Vovk et al., 2005; Tibshirani et al., 2019).

Let $[\boldsymbol{X}] = [X_1, \dots, X_{n_0}, X_{n+1}, \dots, X_k]$ be the unordered set of the features, and fix any value $[\boldsymbol{x}] = [x_1, \dots, x_{n_0}, x_{n_0+1}, \dots, x_{n_0+k}]$. Then, given $[\boldsymbol{X}] = [\boldsymbol{x}]$, the only randomness that remains lies in which value in $\boldsymbol{x}$ corresponds to each $X_i$. Proposition A.4 is a direct implication of Lemma A.2, extending the weighted exchangeability notion in Tibshirani et al. (2019) to multiple test points.

**Proposition A.4.** *Let $\Pi$ be the collection of all permutations of $[n_0 + k]$. Then, for any fixed values $x_1, \dots, x_{n_0}, x_{n_0+1}, \dots, x_{n_0+k}$, and any permutation $\pi \colon [n_0 + k] \to [n_0 + k]$,*

$$\mathbb{P}^{\mathrm{cond}}\Big(X_1 = x_{\pi(1)}, \dots, X_{n_0+k} = x_{\pi(n_0+k)} \,\Big|\, [\boldsymbol{X}] = [\boldsymbol{x}]\Big) = \frac{\bar{w}(\pi; x_{1:n_0+k})}{\sum_{\pi' \in \Pi} \bar{w}(\pi'; x_{1:n_0+k})},$$

*where we define $\bar{w}(\pi; x_{1:n_0+k}) := \prod_{j=1}^{k} w(x_{\pi(n_0+j)})$, and $\mathbb{P}^{\mathrm{cond}}(\cdot)$ denotes the distribution after conditioning on $\{Z_i\}_{i=1}^{n} \cup \{Z_{n+j}\}_{j \ge 1}$ as well as $\max_{1 \le j \le k} Y_{n+j} = 0$.*

Proposition A.4 describes the probability of $(X_1, \dots, X_{n_0+k})$ taking on the value of any permutation of $\boldsymbol{x}$ conditional on the unordered set of their realized values. Such a conditional permutation-style result is foundational for many results in conformal inference, thereby connecting our p-value to existing conformal p-values. Specifically, when $w(\cdot) \equiv 1$ and $m = 1$, it reduces to the standard exchangeability condition, and the p-value (6) coincides with the classical conformal p-value. In general with $m = 1$, it reduces to the weighted exchangeablity studied in (Tibshirani et al., 2019), and our p-value (6) coincides with the weighted conformal p-value (Tibshirani et al., 2019; Jin & Candès, 2023a). We discuss the connections in the next subsection.

## A.3 CONFORMAL P-VALUE FOR ONE TEST POINT

Our p-values build on conformal prediction ideas (Vovk et al., 2005). To provide more contexts, we briefly discuss the widely used conformal p-value for one test point, the key concept in deriving conformal prediction sets. Given any function $s \colon \mathcal{X} \times \mathcal{Y} \to \mathbb{R}$ (often referred to as "nonconformity score") whose training process is independent of the calibration data $\{(X_i, Y_i)\}_{i=1}^{n}$ and a new test point $X_{n+1}$, the conformal p-value is defined as

$$p(y) = \frac{1 + \sum_{i=1}^{n} \mathbb{1}\{s(X_i, Y_i) \le s(X_{n+1}, y)\}}{n+1}. \tag{7}$$

When $\{(X_i, Y_i)\}_{i=1}^{n+1}$ are exchangeable (such as i.i.d.) across $i \in [n+1]$, one can show that $p(Y_{n+1})$ is dominated by $\text{Unif}[0,1]$. This property is key in constructing prediction sets for the unknown $Y_{n+1}$. Recently, conformal p-values based on exchangeability are used in the context of multiple testing for detecting outliers (Bates et al., 2023) and identifying test instances with desirable label values (Jin & Candès, 2023b). Extensions to covariate shift settings are also studied for individual p-value (Tibshirani et al., 2019) and multiple testing (Jin & Candès, 2023a).

Here we briefly discuss Remark 3.2. Note that when $m = 1$, $n := n_0$ and $w(x) \equiv 1$ in (7), setting

$$V(x_1, \ldots, x_{n_0}, x_{n+1}) = s(x_{n+1}, 0),$$

where we use $y = 0$ as a placeholder, our p-value (6) reduces to $p(0)$ in (7).

## A.4 CONNECTION TO OUTLIER DETECTION UNDER COVARIATE SHIFT

A more explicit connection is that of our p-value to the outlier detection p-value. Formally, for a test point $X_{n+1}$ independent of calibration "inliers" $\{X_i\}_{i=1}^n \overset{\text{i.i.d.}}{\sim} P$, Bates et al. (2023) proposes using the conformal p-value (the signs are flipped to be consistent with the current notations)

$$p = \frac{1 + \sum_{i=1}^n \mathbb{1}\{s(X_i) \leq s(X_{n+1})\}}{n+1}$$

to test the null hypothesis $H_0 : X_{n+1} \sim P$, where $s : \mathcal{X} \to \mathbb{R}$ is a score indicating how likely a value $x$ is an outlier compared with the normal values under $P$. Under the null hypothesis $H_0$, it can be shown that the p-value $p$ is dominated by $\text{Unif}[0,1]$.

Jin & Candès (2023a) extend this problem to outlier detection under covariate shift, where $H_0 : X_{n+1} \sim Q$ for some distribution $Q$ obeying $\mathrm{d}Q / \mathrm{d}P(x) = w(x)$ for a known density ratio $w(\cdot)$. The corresponding weighted conformal p-value is

$$p_w = \frac{w(X_{n+1}) + \sum_{i=1}^n w(X_i)\, \mathbb{1}\{s(X_i) \leq s(X_{n+1})\}}{\sum_{i=1}^{n+1} w(X_i)}. \tag{8}$$

In this case, our p-value (6) reduces to (8) by setting

$$V(x_1, \ldots, x_{n_0}, x_{n+1}) = s(x_{n+1}).$$

That is, the score function in our setting can be viewed as an "outlier" score for the new test point.

Again for generality, we use the index $k$ (instead of $N$) to indicate that $\{X_{n+j}\}_{j=1}^k$ might be a subset of the original proposal. Finally, we remark that by Lemma A.2, our null hypotheses $H_k : \max_{1 \leq j \leq k} Y_{n+j} = 0$ implies $X_{n+j} \overset{\text{i.i.d.}}{\sim} Q_0$ where $\mathrm{d}Q_0 / \mathrm{d}P_0(x) = w(x)$ and $\{X_i\}_{i \in \mathcal{I}_0} \overset{\text{i.i.d.}}{\sim} P_0$. Thus, our p-value $p_k$ can be viewed as testing the global null hypotheses

$$\tilde{H}_k : \{X_{n+j}\}_{j=1}^k \overset{\text{i.i.d.}}{\sim} Q_0$$

using calibration "inliers" $\{X_i\}_{i \in \mathcal{I}_0} \overset{\text{i.i.d.}}{\sim} P_0$. Therefore, $p_k$ can be viewed as extending $p_w$ in (8) to simultaneous inference for multiple test points.

## A.5 CHOICE OF SCORE FUNCTION

Drawing inspirations from classical permutation tests, we suggest several natural choices:

- *Max-pooling.* Since we aim to judge whether *any one* of the test samples is promising enough, a natural choice is $V(x_{1:n_0}, x_{n+1:n+k}) = \max_{1 \leq j \leq k} \hat{\mu}(x_{n+j})$, the maximum predicted test score.

- *Sum-of-prediction.* Inspired by Fisher's randomization test in causal inference (Fisher & Fisher, 1971), we can compare the average predicted values between test and calibration data. Up to permutations, it is equivalent to setting $V(x_{1:n_0}, x_{n+1:n+k}) = \sum_{j=1}^k \hat{\mu}(x_{n+j})$.

- *Rank-sum statistic.* Inspired by Wilcoxon's rank-sum test, we compare the sum of ranks of the test points. Up to permutations, we set $V(x_{1:n_0}, x_{n+1:n+k}) = \sum_{j=1}^k R_{n+j}$, where $R_{n+j}$ is the rank of $\hat{\mu}(X_{n+j})$ among $\{\hat{\mu}(X_i)\}_{i=1}^{n_0} \cup \{\hat{\mu}(X_{n+\ell})\}_{\ell=1}^k$ (in an ascending order).

- *Likelihood ratio statistic.* Inspired by the likelihood ratio test, we observe that $\hat{\mu}(x)$ can be viewed as an estimator $\mathbb{P}(Y = 1 \mid X = x)$, thus $\hat{\mu}(x)/(1 - \hat{\mu}(X_{n+j}))$ serves as an estimator for the likelihood ratio for testing $Y_{n+j} = 0$ given $X_{n+j}$. Accordingly, we consider the joint likelihood ratio for the test points and set $V(x_{1:n_0}, x_{n+1:n+k}) = \sum_{j=1}^{k} \log(\hat{\mu}(x_{n+j})/(1 - \hat{\mu}(x_{n+j})))$.

## A.6 NON-SPLIT VERSION OF CONFHIT

In the main text, we discuss CONFHIT when the conformity score is based on a pre-trained model $\hat{\mu}\colon \mathcal{X} \to \mathbb{R}$, as standard in split conformal prediction (Vovk et al., 2005). This makes our framework naturally compatible with any pre-trained prediction models in the literature (such as those in our experiments). However, practitioners may want to involve all the labeled data in both model training and calibration of p-values. In this part, we present an extension of CONFHIT where the labeled inactive data are used in both training and calibration, yet still maintaining type-I error control. Notably, this feature is unique to our problem (since we condition on the labels on the null event).

Formally, we let $\mathcal{A}$ be any generic training algorithm that produces a prediction model $\mathcal{A}(Z_1, \ldots, Z_N)$ based on labeled input data $(Z_1, \ldots, Z_N)$ with $Z_i = (X_i, Y_i)$. We assume that the training process of $\mathcal{A}$ is permutation invariant to its input data, which is satisfied by many commonly used algorithms.

We first consider the certification problem. Recall that the inactive labeled data is $\{(X_i, 0)\}_{i=1}^{n_0}$, the active labeled data, without loss of generality, is $\{(X_i, 1)\}_{i=n_0+1}^{n}$, and the test data is $\{X_{n+j}\}_{j=1}^{k}$. We consider the imputed data $\tilde{Z}_{n+j} = (X_{n+j}, 0)$ for $1 \leq j \leq k$, and

$$\tilde{\mathcal{D}} = (Z_1, \ldots, Z_n, \tilde{Z}_{n+1}, \ldots, \tilde{Z}_{n+k}).$$

Then, we let $\hat{\mu}(\cdot) = \mathcal{A}(\tilde{\mathcal{D}})$, i.e., the trained model with the imputed data. Finally, we define the conformity scores $V$ as well as the p-value in the same way as in Section 3.1. These p-values for each $k \in \mathbb{N}^+$ can then be combined in the same way as in Section 3.2 to address the design problem.

It is straightforward to see that our p-value remains valid conditional on $\max_{1 \leq j \leq k}\{Y_{n+j}\} = 0$: Since the trained model $\hat{\mu}$ is invariant to the permutation of data in $\tilde{\mathcal{D}}$, under any permutation over $\{1, \ldots, n_0, n+1, \ldots, n+k\}$, the trained model remains the same, and the weighted exchangeability between the conformity scores follows exactly the same arguments.

## B TECHNICAL PROOF

### B.1 PROOF OF THEOREM 3.4

*Proof of Theorem 3.4.* We define $K_0 = \max\{k \leq N\colon \max_{1 \leq j \leq k} Y_{n+j} = 0\}$, i.e., the last instance where all earlier molecules are non-positive. We also let $K_0 = 0$ when $Z_{n+1} = 1$ so that $K_0$ is well-defined. Note that $K_0$ is measurable with respect to $\{Y_{n+j}\}_{j \geq 1}$. Conditional on $\{Y_{n+j}\}_{j \geq 1}$, $K_0$ thus becomes deterministic, and by the definition of $\hat{N}$, we have

$$\mathbb{P}\Big( \max_{1 \leq j \leq \hat{N}} Y_{n+j} = 0 \,\Big|\, \{Y_{n+j}\}_{j \geq 1} \Big) = \mathbb{P}\Big( \hat{N} > 0, \max_{j \leq \hat{N}} Z_{n+j} = 0 \,\Big|\, \{Y_{n+j}\}_{j \geq 1} \Big)$$

$$= \mathbb{P}\Big( 0 < \hat{N} \leq K_0 \,\Big|\, \{Z_{n+j}\}_{j \geq 1} \Big) \mathbb{1}\{K_0 > 0\}. \qquad (*)$$

Since the p-values are monotone, we know $\hat{N} \leq K_0$ is equivalent to $p_{K_0} \leq \alpha$, hence

$$(*) = \mathbb{P}\Big( p_{K_0} \leq \alpha \,\Big|\, \{Z_{n+j}\}_{j \geq 1} \Big) \mathbb{1}\{K_0 > 0\}$$

$$= \mathbb{P}\Big( p_{K_0} \leq \alpha \,\Big|\, \{Z_{n+j}\}_{j \geq 1}, \max_{j \leq K_0} Z_{n+j} = 0 \Big) \mathbb{1}\{K_0 > 0\} \leq \alpha.$$

Here the second line uses the fact that $\max_{j \leq K_0} Z_{n+j}$ is measurable with respect to $\{Z_{n+j}\}_{j \geq 1}$, as well as the validity condition (ii). Finally, by the tower property, we obtain the desired result. $\qquad \square$

### B.2 PROOF OF LEMMA A.2

*Proof of Lemma A.2.* Conditional on all calibration and test labels as well as $\max_{1 \leq j \leq k} Y_{n+j} = 0$, it is clear that the features in $\mathcal{I}_0$ are i.i.d. from $P_0 := P_{X \mid Y=0}$, whereas the features $\{X_{n+j}\}_{j=1}^{k}$ are

i.i.d. from $Q_0 := Q_{X\,|\,Y=0}$. By the Bayes' rule, the two distributions are related by

$$\frac{\mathrm{d}Q_0}{\mathrm{d}P_0}(x) = \frac{Q(Y=0\,|\,X=x)}{P(Y=0\,|\,X=x)}\frac{\mathrm{d}Q_X}{\mathrm{d}P_X}(x) = w(x),$$

where $P_X, Q_X$ are the distribution of $X$ under $P$ and $Q$, respectively. Here, we used the fact that $Q(Y=0\,|\,X=x) = P(Y=0\,|\,X=x)$ due to the covariate shift assumption. $\qquad\square$

### B.3 PROOF OF THEOREM A.1

*Proof of Theorem A.1.* Throughout we condition on $\{Z_i\}_{i\geq 1}$ and $\max_{j\leq k} Z_{n+j} = 0$, and let $\mathbb{P}^{\mathrm{cond}}$ denote the distribution conditional on these information. For any fixed vector $\boldsymbol{x} \in \mathcal{X}^{n_0+k}$, conditional on $[\boldsymbol{X}] = [\boldsymbol{x}]$, we let $p_k(\pi;\boldsymbol{x})$ be the p-value obtained when the data is realized as $(X_1,\ldots,X_{n_0},X_{n+1},\ldots,X_{n+k}) = (x_{\pi(1)},\ldots,x_{\pi(n_0)},x_{\pi(n+1)},\ldots,x_{\pi(n+k)})$. We assume $V(\pi;\boldsymbol{X})$'s have no ties almost surely for simplicity, but all the arguments go through in general.

Under the conditions of Lemma A.2, Proposition A.4 implies that

$$\mathbb{P}^{\mathrm{cond}}(p_k = p_k(\pi;\boldsymbol{x})\,|\,[\boldsymbol{X}] = [\boldsymbol{x}])$$
$$= \mathbb{P}^{\mathrm{cond}}(\boldsymbol{X} = (x_{\pi(1)},\ldots,x_{\pi(n+k)})\,|\,[\boldsymbol{X}] = [\boldsymbol{x}]) = \frac{\bar{w}(\pi;\boldsymbol{x})}{\sum_{\pi'\in\Pi}\bar{w}(\pi';\boldsymbol{x})}.$$

Therefore, noting that by definition

$$p_k(\pi;\boldsymbol{x}) = \frac{\sum_{\tilde{\pi}\in\Pi}\bar{w}(\tilde{\pi}\circ\pi;\boldsymbol{x})\,\mathbb{1}\{V(\tilde{\pi}\circ\pi;\boldsymbol{x})\geq V(\pi;\boldsymbol{x})\}}{\sum_{\tilde{\pi}\in\Pi}\bar{w}(\tilde{\pi};\boldsymbol{x})} = \frac{\sum_{\tilde{\pi}\in\Pi}\bar{w}(\tilde{\pi};\boldsymbol{x})\,\mathbb{1}\{V(\tilde{\pi};\boldsymbol{x})\geq V(\pi;\boldsymbol{x})\}}{\sum_{\tilde{\pi}\in\Pi}\bar{w}(\tilde{\pi};\boldsymbol{x})},$$

and letting $\pi_t$ be the permutation with the largest $V(\pi_t;\boldsymbol{x})$ such that $p_k(\pi_t;\boldsymbol{x})\leq t$, we have

$$\mathbb{P}^{\mathrm{cond}}(p_k\leq t\,|\,[\boldsymbol{X}] = [\boldsymbol{x}]) = \sum_{\pi\in\Pi}\mathbb{1}\{p_k(\pi;\boldsymbol{x})\leq t\}\cdot\mathbb{P}^{\mathrm{cond}}(p_k = p_k(\pi;\boldsymbol{x})\,|\,[\boldsymbol{X}] = [\boldsymbol{x}])$$
$$= \sum_{\pi\in\Pi}\mathbb{1}\{p_k(\pi;\boldsymbol{x})\leq t\}\cdot\frac{\bar{w}(\pi;\boldsymbol{x})}{\sum_{\pi'\in\Pi}\bar{w}(\pi';\boldsymbol{x})}$$
$$= \sum_{\pi\in\Pi}\mathbb{1}\{V(\pi;\boldsymbol{x})\geq V(\pi_t;\boldsymbol{x})\}\cdot\frac{\bar{w}(\pi;\boldsymbol{x})}{\sum_{\pi'\in\Pi}\bar{w}(\pi';\boldsymbol{x})} = p_k(\pi_t;\boldsymbol{x})\leq t.$$

Here the first equality uses the conditional distribution described above, the third equality uses the monotonicity of $p_k(\pi;\boldsymbol{x})$ in $V(\pi;\boldsymbol{x})$ and the definition of $\pi_t$, and the last equality holds by definition. Marginalizing over $[\boldsymbol{X}]$ (but still conditional on $\{Z_i\}_{i=1}^{n+k}$ and $\max_{1\leq j\leq k} Y_{n+j} = 0$) then leads to the desired result. $\qquad\square$

### B.4 PROOF OF THEOREM 3.1

*Proof of Theorem 3.1.* As usual, throughout, we condition on $\{Y_i\}_{i\geq 1}$ and $\max_{1\leq j\leq k} Y_{n+j} = 0$, and use $\mathbb{P}^{\mathrm{cond}}$ to denote the distribution given these information. We condition on the event that the unordered set $[\boldsymbol{x}] = [\boldsymbol{x}]$ for any fixed vector $\boldsymbol{x} = (x_1,\ldots,x_{n_0},x_{n+1},\ldots,x_{n+k})$. The labels $\{Z_i\}_{i=1}^{n+k}$ are also conditioned on as usual. The randomness then comes from (i) which values in $[\boldsymbol{x}]$ correspond to each data point, and (ii) the randomly sampled permutations.

Let $\hat{\pi}$ be the unique permutation of $\{1,\ldots,n_0,n+1,\ldots,n+k\}$ such that $\boldsymbol{X} = (x_{\hat{\pi}(1)},\ldots,x_{\hat{\pi}(n+k)})$. Then for any permutation $\pi\in\Pi$, we have

$$V(\pi;\boldsymbol{X}) = V(\pi\circ\pi^{(0)};\boldsymbol{X}) = V(\pi\circ\hat{\pi};\boldsymbol{x}),$$

and similarly for $\bar{w}(\pi;\boldsymbol{X}) = \bar{w}(\pi\circ\hat{\pi};\boldsymbol{x})$. By Proposition A.4, we know that $\mathbb{P}(\hat{\pi} = \pi\,|\,[\boldsymbol{X}] = [\boldsymbol{x}]) \propto \bar{w}(\pi;\boldsymbol{x})$, where we recall the definition of $\bar{w}(\pi;\boldsymbol{x}) = \prod_{j=1}^k w(x_{\pi(n+j)})$. In addition, we have

$$p_k^{\mathrm{rand}} = \frac{\bar{w}(\hat{\pi};\boldsymbol{x}) + \sum_{b=1}^B \bar{w}(\hat{\pi}\circ\pi^{(b)};\boldsymbol{x})\,\mathbb{1}\{V(\hat{\pi};\boldsymbol{x})\leq V(\hat{\pi}\circ\pi^{(b)};\boldsymbol{x})\}}{w(\hat{\pi};\boldsymbol{x}) + \sum_{b=1}^B w(\hat{\pi}\circ\pi^{(b)};\boldsymbol{x})},$$

where $\boldsymbol{x}$ is the fixed vector, and the randomness comes from $\hat{\pi}$ and $\{\pi^{(b)}\}_{b=1}^B$.

For notational simplicity, we denote $\hat{\pi}^{(0)} = \hat{\pi}$, and $\hat{\pi}^{(b)} = \hat{\pi} \circ \pi^{(b)}$ for $b = 1, \ldots, B$. We now study the joint distribution of $(\hat{\pi}^{(0)}, \hat{\pi}^{(1)}, \ldots, \hat{\pi}^{(B)})$. For any fixed permutations $\pi_0, \ldots, \pi_B \in \Pi$, we note

$$\mathbb{P}^{\text{cond}}\left(\hat{\pi}^{(0)} = \pi_0, \hat{\pi}^{(1)} = \pi_1, \ldots, \hat{\pi}^{(B)} = \pi_B \,\Big|\, [\boldsymbol{X}] = [\boldsymbol{x}]\right)$$

$$= \mathbb{P}^{\text{cond}}\left(\hat{\pi} = \pi_0, \pi^{(1)} = \pi_0^{-1} \circ \pi_1, \ldots, \hat{\pi}^{(B)} = \pi_0^{-1} \circ \pi_B \,\Big|\, [\boldsymbol{X}] = [\boldsymbol{x}]\right)$$

$$= \frac{1}{|\Pi|^B} \cdot \mathbb{P}^{\text{cond}}(\hat{\pi} = \pi_0 \,|\, [\boldsymbol{X}] = [\boldsymbol{x}]) \propto \bar{w}(\pi_0; \boldsymbol{x}),$$

where $\pi_0^{-1}$ is the inverse permutation of $\pi_0$. Above, the last equality uses the fact that $\pi^{(1)}, \ldots, \pi^{(B)}$ are i.i.d. uniformly sampled from $\Pi$ and independent of everything. Therefore, when we further conditional on the unordered set of realized values of the permutations $[\hat{\pi}^{(0)}, \hat{\pi}^{(1)}, \ldots, \hat{\pi}^{(B)}] = [\pi_0, \ldots, \pi_B]$ for any (fixed) value of $\pi_0, \ldots, \pi_B$, we know that

$$\mathbb{P}^{\text{cond}}\left((\hat{\pi}^{(0)}, \hat{\pi}^{(1)}, \ldots, \hat{\pi}^{(B)}) = (\pi_{\sigma(0)}, \ldots, \pi_{\sigma(B)}) \,\Big|\, [\hat{\pi}^{(0)}, \hat{\pi}^{(1)}, \ldots, \hat{\pi}^{(B)}] = [\pi_0, \ldots, \pi_B], \; [\boldsymbol{X}] = [\boldsymbol{x}]\right)$$

$$= \frac{\bar{w}(\pi_{\sigma(0)}; \boldsymbol{x})}{\sum_{\sigma \in \tilde{\Pi}} \bar{w}(\pi_{\sigma(0)}; \boldsymbol{x})},$$

where $\tilde{\Pi}$ is the collection of all permutations of $\{0, 1, \ldots, B\}$. This also means that for every $b \in \{0, 1, \ldots, B\}$,

$$\mathbb{P}^{\text{cond}}\left(\hat{\pi}^{(0)} = \pi_b \,\Big|\, [\hat{\pi}^{(0)}, \hat{\pi}^{(1)}, \ldots, \hat{\pi}^{(B)}] = [\pi_0, \ldots, \pi_B]\right) = \frac{\bar{w}(\pi_b; \boldsymbol{x})}{\sum_{b'=0}^B \bar{w}(\pi_{b'}; \boldsymbol{x})}. \tag{9}$$

On the other hand, for any $\sigma \in \tilde{\Pi}$, when $(\hat{\pi}^{(0)}, \hat{\pi}^{(1)}, \ldots, \hat{\pi}^{(B)}) = (\pi_{\sigma(0)}, \ldots, \pi_{\sigma(B)})$, we have

$$p_k^{\text{cond}} = \frac{\bar{w}(\pi_{\sigma(0)}; \boldsymbol{x}) + \sum_{b=1}^B \bar{w}(\pi_{\sigma(b)}; \boldsymbol{x}) \mathbb{1}\{V(\pi_{\sigma(0)}; \boldsymbol{x}) \leq V(\pi_{\sigma(b)}; \boldsymbol{x})\}}{\bar{w}(\pi_{\sigma(0)}; \boldsymbol{x}) + \sum_{b=1}^B \bar{w}(\pi_{\sigma(b)}; \boldsymbol{x})}$$

$$= \frac{\sum_{b=0}^B \bar{w}(\pi_b; \boldsymbol{x}) \mathbb{1}\{V(\pi_{\sigma(0)}; \boldsymbol{x}) \leq V(\pi_b; \boldsymbol{x})\}}{\sum_{b=0}^B \bar{w}(\pi_b; \boldsymbol{x})}. \tag{10}$$

That is, the value of $p_k^{\text{cond}}$ only depends on which value among $(\pi_0, \ldots, \pi_B)$ the data permutation $\hat{\pi}^{(0)}$ takes on. Thus,

$$\mathbb{P}^{\text{cond}}\left(p_k^{\text{rand}} \leq t \,\Big|\, [\hat{\pi}^{(0)}, \hat{\pi}^{(1)}, \ldots, \hat{\pi}^{(B)}] = [\pi_0, \ldots, \pi_B], \; [\boldsymbol{X}] = [\boldsymbol{x}]\right)$$

$$= \sum_{\sigma \in \tilde{\Pi}} \mathbb{P}\left((\hat{\pi}^{(0)}, \hat{\pi}^{(1)}, \ldots, \hat{\pi}^{(B)}) = (\pi_{\sigma(0)}, \ldots, \pi_{\sigma(B)}) \,\Big|\, [\hat{\pi}^{(0)}, \hat{\pi}^{(1)}, \ldots, \hat{\pi}^{(B)}] = [\pi_0, \ldots, \pi_B]\right)$$

$$\times \mathbb{1}\left\{\frac{\sum_{b=0}^B \bar{w}(\pi_b; \boldsymbol{x}) \mathbb{1}\{V(\pi_{\sigma(0)}; \boldsymbol{x}) \leq V(\pi_b; \boldsymbol{x})\}}{\sum_{b=0}^B \bar{w}(\pi_b; \boldsymbol{x})} \leq t\right\}$$

$$= \sum_{b^*=0}^B \mathbb{P}\left(\hat{\pi}^{(0)} = \pi_{b^*} \,\Big|\, [\hat{\pi}^{(0)}, \hat{\pi}^{(1)}, \ldots, \hat{\pi}^{(B)}] = [\pi_0, \ldots, \pi_B]\right)$$

$$\times \mathbb{1}\left\{\frac{\sum_{b=0}^B \bar{w}(\pi_b; \boldsymbol{x}) \mathbb{1}\{V(\pi_{b^*}; \boldsymbol{x}) \leq V(\pi_b; \boldsymbol{x})\}}{\sum_{b=0}^B \bar{w}(\pi_b; \boldsymbol{x})} \leq t\right\}$$

$$= \sum_{b^*=0}^B \frac{\bar{w}(\pi_{b^*}; \boldsymbol{x})}{\sum_{b'=0}^B \bar{w}(\pi_{b'}; \boldsymbol{x})} \mathbb{1}\left\{\frac{\sum_{b=0}^B \bar{w}(\pi_b; \boldsymbol{x}) \mathbb{1}\{V(\pi_{b^*}; \boldsymbol{x}) \leq V(\pi_b; \boldsymbol{x})\}}{\sum_{b=0}^B \bar{w}(\pi_b; \boldsymbol{x})} \leq t\right\} \leq t.$$

Here, the first equality follows from (10), the third equality follows from (9), and the last inequality is by definition. Finally, marginalizing over $[\boldsymbol{X}]$ and $[\hat{\pi}^{(0)}, \ldots, \hat{\pi}^{(B)}]$ yields the desired result. $\qquad \square$

### B.5    Proof of Theorem 3.5

In this part, we prove the robustness of joint weighted conformal p-value with estimated density ratio. For generality, we use a general index $k$ instead of the total budget $N$, to indicate that $\{X_{n+j}\}_{j=1}^{k}$ can be a subset of the original samples (as in Section 3.2). For conceptual simplicity, we present the detailed proof for deterministic p-values (Appendix A.1), and the result for randomized p-values in the main text naturally follows with the same ideas.

*Proof of Theorem 3.5.* Let $\mathbb{P}^{\mathrm{cond}}(\cdot)$ denote the distribution conditional on all the labels $\{Y_i\}_{i\geq 1}$ and the event $\max_{1\leq j\leq k} Y_{n+j} = 0$. Following the arguments in the proof of Theorem A.1, for any fixed value $\boldsymbol{x}$, we let $p_k(\pi; \boldsymbol{x})$ be the p-value with the correct weights $w(\cdot)$ obtained when the data is realized as $(X_1, \ldots, X_{n_0}, X_{n+1}, \ldots, X_{n+k}) = (x_{\pi(1)}, \ldots, x_{\pi(n_0)}, x_{\pi(n+1)}, \ldots, x_{\pi(n+k)})$, and similarly we define $\hat{p}_k(\pi; \boldsymbol{x})$. Then we have $\hat{p}_k(V(\pi; \boldsymbol{x})) = \hat{p}_k(\pi; \boldsymbol{x})$ where $\hat{p}_k(v)$ is defined in Theorem 3.5, and by definition, we have

$$\sum_{\pi \in \Pi} \mathbb{1}\{\hat{p}_k(\pi; \boldsymbol{x}) \leq \hat{t}\} \cdot \frac{\hat{\bar{w}}(\pi; \boldsymbol{x})}{\sum_{\pi' \in \Pi} \hat{\bar{w}}(\pi'; \boldsymbol{x})} = \hat{p}_k(\hat{v}) \leq \hat{t} \leq t.$$

In addition, by the definition of $\hat{t}$, we have $\hat{p}_k(\pi; \boldsymbol{x}) \leq \hat{t}$ if and only if $\hat{p}_k(\pi; \boldsymbol{x}) \leq t$ for any $\pi \in \Pi$. As a result, we have

$$\mathbb{P}^{\mathrm{cond}}\big(\hat{p}_k \leq t \,\big|\, [\boldsymbol{X} = \boldsymbol{x}]\big) = \sum_{\pi \in \Pi} \mathbb{1}\{\hat{p}_k(\pi; \boldsymbol{x}) \leq t\} \cdot \frac{\bar{w}(\pi; \boldsymbol{x})}{\sum_{\pi' \in \Pi} \bar{w}(\pi'; \boldsymbol{x})}$$

$$= \sum_{\pi \in \Pi} \mathbb{1}\{\hat{p}_k(\pi; \boldsymbol{x}) \leq \hat{t}\} \cdot \frac{\bar{w}(\pi; \boldsymbol{x})}{\sum_{\pi' \in \Pi} \bar{w}(\pi'; \boldsymbol{x})}$$

$$\leq \hat{t} + \sum_{\pi \in \Pi} \mathbb{1}\{\hat{p}_k(\pi; \boldsymbol{x}) \leq \hat{t}\} \left( \frac{\bar{w}(\pi; \boldsymbol{x})}{\sum_{\pi' \in \Pi} \bar{w}(\pi'; \boldsymbol{x})} - \frac{\hat{\bar{w}}(\pi; \boldsymbol{x})}{\sum_{\pi' \in \Pi} \hat{\bar{w}}(\pi'; \boldsymbol{x})} \right)$$

$$\leq \hat{t} + \sum_{\pi \in \Pi} \mathbb{1}\{V(\pi; \boldsymbol{x}) \geq \hat{v}\} \left( \frac{\bar{w}(\pi; \boldsymbol{x})}{\sum_{\pi' \in \Pi} \bar{w}(\pi'; \boldsymbol{x})} - \frac{\hat{\bar{w}}(\pi; \boldsymbol{x})}{\sum_{\pi' \in \Pi} \hat{\bar{w}}(\pi'; \boldsymbol{x})} \right)$$

$$\leq t + \frac{\sum_{\pi \in \Pi} \bar{w}(\pi; \boldsymbol{x}) \mathbb{1}\{V(\pi; \boldsymbol{x}) \geq \hat{v}\}}{\sum_{\pi' \in \Pi} \bar{w}(\pi'; \boldsymbol{x})} - \frac{\sum_{\pi \in \Pi} \hat{\bar{w}}(\pi; \boldsymbol{x}) \mathbb{1}\{V(\pi; \boldsymbol{x}) \geq \hat{v}\}}{\sum_{\pi' \in \Pi} \hat{\bar{w}}(\pi'; \boldsymbol{x})} =: t + \hat{\Delta}.$$

Rearranging the terms, we have

$$\hat{\Delta} = \frac{\big(\sum_{\pi \in \Pi} \bar{w}(\pi; \boldsymbol{x}) \mathbb{1}\{V(\pi; \boldsymbol{x}) \geq \hat{v}\}\big)\big(\sum_{\pi \in \Pi} \hat{\bar{w}}(\pi; \boldsymbol{x}) \mathbb{1}\{V(\pi; \boldsymbol{x}) < \hat{v}\}\big)}{\big(\sum_{\pi' \in \Pi} \bar{w}(\pi'; \boldsymbol{x})\big)\big(\sum_{\pi' \in \Pi} \hat{\bar{w}}(\pi'; \boldsymbol{x})\big)}$$

$$- \frac{\big(\sum_{\pi \in \Pi} \bar{w}(\pi; \boldsymbol{x}) \mathbb{1}\{V(\pi; \boldsymbol{x}) < \hat{v}\}\big)\big(\sum_{\pi \in \Pi} \hat{\bar{w}}(\pi; \boldsymbol{x}) \mathbb{1}\{V(\pi; \boldsymbol{x}) \geq \hat{v}\}\big)}{\big(\sum_{\pi' \in \Pi} \bar{w}(\pi'; \boldsymbol{x})\big)\big(\sum_{\pi' \in \Pi} \hat{\bar{w}}(\pi'; \boldsymbol{x})\big)}$$

$$\leq \frac{\big(\sum_{\pi \in \Pi} [\hat{\bar{w}}(\pi; \boldsymbol{x}) + \hat{\delta}^{-}(\pi; \boldsymbol{x})] \mathbb{1}\{V(\pi; \boldsymbol{x}) \geq \hat{v}\}\big)\big(\sum_{\pi \in \Pi} \hat{\bar{w}}(\pi; \boldsymbol{x}) \mathbb{1}\{V(\pi; \boldsymbol{x}) < \hat{v}\}\big)}{\big(\sum_{\pi' \in \Pi} \bar{w}(\pi'; \boldsymbol{x})\big)\big(\sum_{\pi' \in \Pi} \hat{\bar{w}}(\pi'; \boldsymbol{x})\big)}$$

$$- \frac{\big(\sum_{\pi \in \Pi} [\hat{\bar{w}}(\pi; \boldsymbol{x}) - \hat{\delta}^{+}(\pi; \boldsymbol{x})] \mathbb{1}\{V(\pi; \boldsymbol{x}) < \hat{v}\}\big)\big(\sum_{\pi \in \Pi} \hat{\bar{w}}(\pi; \boldsymbol{x}) \mathbb{1}\{V(\pi; \boldsymbol{x}) \geq \hat{v}\}\big)}{\big(\sum_{\pi' \in \Pi} \bar{w}(\pi'; \boldsymbol{x})\big)\big(\sum_{\pi' \in \Pi} \hat{\bar{w}}(\pi'; \boldsymbol{x})\big)}$$

$$\leq \frac{\big(\sum_{\pi \in \Pi} \hat{\delta}^{-}(\pi; \boldsymbol{x}) \mathbb{1}\{V(\pi; \boldsymbol{x}) \geq \hat{v}\}\big)\big(\sum_{\pi \in \Pi} \hat{\bar{w}}(\pi; \boldsymbol{x}) \mathbb{1}\{V(\pi; \boldsymbol{x}) < \hat{v}\}\big)}{\big(\sum_{\pi' \in \Pi} \bar{w}(\pi'; \boldsymbol{x})\big)\big(\sum_{\pi' \in \Pi} \hat{\bar{w}}(\pi'; \boldsymbol{x})\big)}$$

$$+ \frac{\big(\sum_{\pi \in \Pi} \hat{\delta}^{+}(\pi; \boldsymbol{x}) \mathbb{1}\{V(\pi; \boldsymbol{x}) < \hat{v}\}\big)\big(\sum_{\pi \in \Pi} \hat{\bar{w}}(\pi; \boldsymbol{x}) \mathbb{1}\{V(\pi; \boldsymbol{x}) \geq \hat{v}\}\big)}{\big(\sum_{\pi' \in \Pi} \bar{w}(\pi'; \boldsymbol{x})\big)\big(\sum_{\pi' \in \Pi} \hat{\bar{w}}(\pi'; \boldsymbol{x})\big)},$$

where we used the non-negativity of the weights and the definition of $\hat{\delta}^{\pm}(\pi; \boldsymbol{x})$. Further, noting that $\hat{t} = \frac{\sum_{\pi \in \Pi} \hat{\bar{w}}(\pi; \boldsymbol{X}) \mathbb{1}\{\hat{v} \leq V(\pi; \boldsymbol{X})\}}{\sum_{\pi \in \Pi} \hat{\bar{w}}(\pi; \boldsymbol{X})}$, we thus have

$$\hat{\Delta} \leq \frac{(1 - \hat{t}) \cdot \big(\sum_{\pi \in \Pi} \hat{\delta}^{-}(\pi; \boldsymbol{x}) \mathbb{1}\{V(\pi; \boldsymbol{x}) \geq \hat{v}\}\big) + \hat{t} \cdot \big(\sum_{\pi \in \Pi} \hat{\delta}^{+}(\pi; \boldsymbol{x}) \mathbb{1}\{V(\pi; \boldsymbol{x}) < \hat{v}\}\big)}{\sum_{\pi' \in \Pi} \bar{w}(\pi'; \boldsymbol{x})},$$

which completes our proof for the results of $p_k$.

Finally, all the arguments still go through (following some arguments in the proof of Theorem 3.1) for $\hat{p}_k$ after replacing $\Pi$ by $\{\pi^{(b)}\}_{b=0}^{B}$ and additionally conditioning on the unordered set of $\{\hat{\pi}, \pi^{(1)}, \ldots, \pi^{(B)}\}$, where $\hat{\pi}$ is the permutation such that $\boldsymbol{X} = (x_{\pi(1)}, \ldots, x_{\pi(n+k)})$. We omit the details here for brevity. □

## C   EXPERIMENTAL SETTING DETAILS

### C.1   CONSTRAINED MOLECULE OPTIMIZATION

For training the generator, we used the publicly available training code from the HGRAPH2GRAPH (Jin et al., 2020) repository. We retained the default training/test splits and randomly held out 20% of the training set for calibration. The test set contains 800 samples and is also sourced from the same repository.

For the SELF-EDIT (Jiao et al., 2023) baseline, we used the same data splits and obtained the training code from its official repository. Both models were trained with their default configurations, without modification.

In all cases, we sampled up to 50 candidate molecules per input until a sufficient number of *eligible* candidates were obtained. We defined *ineligible* candidates as those that were invalid (e.g., chemically invalid), duplicates, or out-of-distribution (OOD) with respect to the training data. We retained only those test samples for which at least $N \in \{5, 10\}$ valid candidates could be generated.

The property oracles were sourced from RDKit for QED and from the Therapeutics Data Commons library for DRD2.

**Density Estimation and Scoring Function.** We trained a molecular property predictor using the Chemprop(Yang et al., 2019) library for both QED and DRD2 tasks. The model used a 3-layer Message passing Neural Network with a final hidden dimension of 4 for feature extraction and was trained for 5 epochs using the default Chemprop trainer settings. We used the `MulticomponentMessagePassing` module without any modification.

For density estimation, we followed the CODRUG approach by extracting penultimate layer features and fitting a Gaussian kernel density estimator (KDE) using the `scipy` library with a bandwidth of 1. To filter OOD samples, we computed the 95th percentile density on the calibration set and removed test-time generated molecules falling below this threshold. Additional samples were drawn until the target number $N$ of valid candidates was met.

For sequential p-value computation, we used 2000 Monte Carlo samples per test instance.

### C.2   HYPERPARAMETER CHOICES AND DESIGN JUSTIFICATIONS

**Density ratio model and feature selection.** For density estimation, we follow the CODRUG procedure, which has been shown to be robust for small-molecule density estimation under covariate shift. We use the penultimate-layer representation of the property predictor, which yields a 4-dimensional feature vector that is expressive yet low-dimensional enough to enable stable kernel density estimation. For bandwidth selection, we adopt the same strategy as CODRUG: we perform a 5-fold cross-validation over bandwidths $\{0.1, 1, 10\}$, compute the KDE likelihood on each held-out split, and choose the bandwidth with the highest average likelihood. This avoids post-hoc tuning and ensures that the KDE matches the calibration distribution as closely as possible.

**Predictive model selection.** Conformal validity does not depend on the complexity of the predictive model, but the power of the method improves when better predictors and more discriminative score functions are used. To avoid unnecessary tuning, we select models with strong reported performance and publicly available implementations. For the CMO tasks (QED/DRD2), we train a standard Chemprop MPNN. For the SBDD task, we use the default EGNN predictor from the TARGETDIFF implementation. These choices are simple, stable, and sufficient for producing informative conformity scores.

**Choice of test statistic.** Since the procedure operates on a *set* of generated molecules, the choice of score statistic affects the power but not the validity. We use the **maximum** score across the set, which is a natural heuristic in this context: if at least one strong candidate is present, the max-score captures this evidence immediately, whereas statistics such as the mean or min dilute the signal with weaker candidates. Empirically, we found the max statistic to produce higher power and lower rejection rates than alternatives.

**Number of permutations ($B$).** We set $B = 2000$ Monte Carlo permutations, which we found sufficient for stable performance while maintaining modest computational cost. Running the full CONFHIT procedure with $B = 2000$ takes only **4 minutes for 800 samples** (see Fig. 6). We also evaluated $B \in \{500, 1000, 2000, 5000\}$ and observed negligible variance in empirical coverage across these values, confirming that the Monte Carlo approximation converges quickly. Runtime scales linearly with $B$, and 2000 provides an effective balance between statistical stability and efficiency.

Figure 6: Error rate as a function of the number of Monte Carlo permutations B for DRD2 with HGraph2Graph (N = 5). The empirical error rate remains stable with minimal variance.

## C.3 STRUCTURE-BASED DRUG DISCOVERY

For this task, we used a pre-trained generative model checkpoint obtained from the official TARGET-DIFF (Guan et al., 2023) repository, along with their corresponding test set.

To estimate scores and extract features, we trained an EGNN-based model using their provided codebase. We modified the final hidden layer to output an 8-dimensional embedding. For validation data, we used protein-ligand complexes from the CrossDocked2020 (Francoeur et al., 2020a) dataset, excluding all training and test set entries to avoid data leakage.

Density estimation was performed as above, using Gaussian KDE with a bandwidth of 1.

## C.4 COMPUTE RESOURCES

All experiments were conducted on a single NVIDIA A100 GPU. The approximate runtime per component is as follows:

- Property predictor training (QED/DRD2): 20 minutes
- HGRAPH2GRAPH generator training: 10 hours
- SELF-EDIT training: 48 hours
- Density estimator training for SBDD: 24 hours
- Inference using TARGETDIFF: 30 minutes
- CONFHIT procedure per test batch: 30 minutes (over 800 samples; $\approx$2.25 seconds/sample)
  - Candidate generation (CMO task): 14 minutes ($\approx$1.05 seconds/sample)

- Predictor feature extraction: 6 minutes ($\approx$0.45 seconds/sample)
- KDE computation: 3 minutes ($\approx$0.23 seconds/sample)
- Running the conformal procedure: 4 minutes ($\approx$0.30 seconds/sample)

### C.5 CODE AND DATA AVAILABILITY

We include with the code the following:

- Model checkpoints.
- Extracted features and CrossDocked identifiers used for calibration in SBDD.
- Calibration and test splits.
- Configuration files for reproducing results across datasets and models.
- Installation instructions.

The code is publicly available at https://github.com/siddharthal/CONFHIT-Conformal-Generative-Design-with-Oracle-Free-Guarantees and is distributed under MIT license.

## D WORKING UNDER AN OVERALL BUDGET

In Section 4.6, we discussed the application of CONFHIT in providing practically useful heuristics while constructing predicting sets when a fixed total budget $B$ is available. The detailed algorithm is provided in Algorithm 2.

---

**Algorithm 2** Working under an overall budget

---

**Input:** Calibration data $\{X_i\}_{i=1}^{n_0}$ with $Y_i = 0$, test batches $\{X^{(t)}\}_{t=1}^T$ of size $N$, conformity score $V$, weight function $\hat{w}$, grid of confidence levels $\mathcal{A} \subset (0,1)$, total budget $B \in \mathbb{N}^+$.

1: Initialize $\hat{P}_{\max} \leftarrow 0, \mathcal{C}^\star \leftarrow \emptyset, \alpha^\star \leftarrow 0$.
2: **for** each $\alpha \in \mathcal{A}$ **do**
3:     For each test batch $t = 1, \ldots, T$, run Algorithm 1 with level $\alpha$ to obtain prunned prediction sets $\mathcal{C}_t^{(\alpha)}$.
4:     Compute total cost $C = \sum_{t=1}^T |\mathcal{C}_t^{(\alpha)}|$.
5:     Record number of empty sets $E = |\{t : \mathcal{C}_t^{(\alpha)} = \emptyset\}|$.
6:     **if** $C > B$ **then**
7:         Sort $\mathcal{C}_t^{(\alpha)}$ in descending order of size.
8:         Iteratively delete sets until total cost $C \leq B$.
9:         Let $D$ be the number of deleted sets.
10:     **else**
11:         Set $D \leftarrow 0$.
12:     **end if**
13:     Estimate positives as $\hat{P}(\alpha) = (1 - \alpha) - E - D$.
14:     **if** $\hat{P}(\alpha) > \hat{P}_{\max}$ **then**
15:         Update $\hat{P}_{\max} \leftarrow \hat{P}(\alpha), \mathcal{C}^\star \leftarrow \{\mathcal{C}_t^{(\alpha)}\}, \alpha^\star \leftarrow \alpha$.
16:     **end if**
17: **end for**
**Output:** Best estimated positives $\hat{P}_{\max}$ with corresponding sets $\mathcal{C}^\star$ at significance level $\alpha^\star$.

---

## E ADDITIONAL EXPERIMENTS

### E.1 DETAILED RESULTS

In this section, we provide error bars for the plots in Sections 4.3 and 4.4. The tables include Error Rate, fraction certified, empty set fraction and mean set size depicted in 4.3 and 4.4 along with standard deviation computed over 5 random runs, across different methods. Further, we report metrics on the Constrained Molecule Optimization (CMO) task, using the Qualitative Estimate of Drug-Likeness (QED) property with success ($Y = 1$) if $\text{QED}(x) > 0.9$, following Jin et al. (2020).

### E.1.1 CERTIFICATION RESULTS

Table 1: DRD2_HGRAPH Certification Results

| $N$ | $\alpha$ | Error Rate (%) | Certified Fraction (%) |
|---|---|---|---|
| 3 | 0.1 | $6.5 \pm 1.1$ | $22.6 \pm 2.4$ |
| 3 | 0.2 | $9.6 \pm 1.4$ | $33.2 \pm 4.1$ |
| 3 | 0.3 | $18.5 \pm 1.9$ | $39.8 \pm 4.7$ |
| 3 | 0.4 | $26.8 \pm 2.1$ | $47.6 \pm 3.5$ |
| 3 | 0.5 | $26.0 \pm 2.0$ | $57.5 \pm 6.8$ |
| 5 | 0.1 | $7.3 \pm 2.0$ | $43.0 \pm 4.3$ |
| 5 | 0.2 | $9.3 \pm 1.2$ | $53.0 \pm 4.6$ |
| 5 | 0.3 | $12.2 \pm 3.2$ | $68.8 \pm 6.2$ |
| 5 | 0.4 | $21.8 \pm 2.6$ | $65.1 \pm 4.6$ |
| 5 | 0.5 | $21.6 \pm 2.2$ | $70.2 \pm 6.1$ |
| 7 | 0.1 | $5.5 \pm 0.6$ | $51.5 \pm 5.1$ |
| 7 | 0.2 | $8.1 \pm 1.2$ | $60.6 \pm 2.5$ |
| 7 | 0.3 | $11.8 \pm 3.1$ | $66.2 \pm 5.2$ |
| 7 | 0.4 | $19.7 \pm 3.2$ | $70.4 \pm 3.6$ |
| 7 | 0.5 | $20.6 \pm 2.2$ | $74.5 \pm 6.2$ |

Table 2: DRD2_SELFEDIT Certification Results

| $N$ | $\alpha$ | Error Rate (%) | Certified Fraction (%) |
|---|---|---|---|
| 3 | 0.1 | $7.1 \pm 0.4$ | $19.1 \pm 2.1$ |
| 3 | 0.2 | $8.7 \pm 1.7$ | $21.5 \pm 3.1$ |
| 3 | 0.3 | $19.8 \pm 1.2$ | $28.2 \pm 3.2$ |
| 3 | 0.4 | $19.8 \pm 1.6$ | $39.3 \pm 4.1$ |
| 3 | 0.5 | $28.3 \pm 3.3$ | $42.5 \pm 5.2$ |
| 5 | 0.1 | $7.0 \pm 1.0$ | $22.8 \pm 3.1$ |
| 5 | 0.2 | $7.4 \pm 1.3$ | $25.9 \pm 2.5$ |
| 5 | 0.3 | $16.4 \pm 2.2$ | $30.2 \pm 4.1$ |
| 5 | 0.4 | $17.4 \pm 2.6$ | $41.7 \pm 4.6$ |
| 5 | 0.5 | $26.9 \pm 3.2$ | $45.7 \pm 5.1$ |
| 7 | 0.1 | $6.3 \pm 0.9$ | $29.0 \pm 2.2$ |
| 7 | 0.2 | $9.8 \pm 1.6$ | $38.3 \pm 3.1$ |
| 7 | 0.3 | $16.8 \pm 2.1$ | $42.3 \pm 4.7$ |
| 7 | 0.4 | $17.1 \pm 2.2$ | $47.6 \pm 4.9$ |
| 7 | 0.5 | $26.4 \pm 4.2$ | $51.6 \pm 5.6$ |

Table 3: QED_HGRAPH Certification Results

| $N$ | $\alpha$ | Error Rate (%) | Certified Fraction (%) |
|---|---|---|---|
| 3 | 0.1 | $7.8 \pm 1.2$ | $49.3 \pm 2.0$ |
| 3 | 0.2 | $13.4 \pm 1.6$ | $47.8 \pm 1.8$ |
| 3 | 0.3 | $16.3 \pm 2.1$ | $54.0 \pm 1.0$ |
| 3 | 0.4 | $19.4 \pm 2.7$ | $56.8 \pm 1.0$ |
| 3 | 0.5 | $21.8 \pm 3.2$ | $60.8 \pm 3.4$ |
| 7 | 0.1 | $7.9 \pm 0.5$ | $52.2 \pm 3.2$ |
| 7 | 0.2 | $14.6 \pm 1.6$ | $56.5 \pm 2.0$ |
| 7 | 0.3 | $17.9 \pm 2.1$ | $56.7 \pm 2.2$ |
| 7 | 0.4 | $20.3 \pm 1.2$ | $65.5 \pm 3.1$ |
| 7 | 0.5 | $22.5 \pm 3.4$ | $72.0 \pm 2.7$ |

Table 4: QED_SELFEDIT Certification Results

| $N$ | $\alpha$ | Error Rate (%) | Certified Fraction (%) |
|---|---|---|---|
| 3 | 0.1 | $7.6 \pm 1.2$ | $42.6 \pm 6.1$ |
| 3 | 0.2 | $10.4 \pm 0.6$ | $54.6 \pm 3.9$ |
| 3 | 0.3 | $14.2 \pm 1.3$ | $52.2 \pm 2.7$ |
| 3 | 0.4 | $15.7 \pm 2.1$ | $57.0 \pm 3.6$ |
| 3 | 0.5 | $17.2 \pm 3.7$ | $62.7 \pm 4.1$ |
| 7 | 0.1 | $11.9 \pm 1.4$ | $52.2 \pm 3.5$ |
| 7 | 0.2 | $21.6 \pm 2.1$ | $62.3 \pm 4.0$ |
| 7 | 0.3 | $29.7 \pm 2.5$ | $61.2 \pm 2.9$ |
| 7 | 0.4 | $30.8 \pm 1.2$ | $77.9 \pm 4.3$ |
| 7 | 0.5 | $32.9 \pm 3.2$ | $78.4 \pm 1.6$ |

Table 5: TARGETDIFF Certification Results

| $N$ | $\alpha$ | Error Rate (%) | Certified Fraction (%) |
|---|---|---|---|
| 5 | 0.1 | $8.4 \pm 2.6$ | $42.9 \pm 2.5$ |
| 5 | 0.2 | $10.8 \pm 3.0$ | $52.1 \pm 2.9$ |
| 5 | 0.3 | $11.4 \pm 2.3$ | $57.7 \pm 3.8$ |
| 5 | 0.4 | $14.9 \pm 4.4$ | $64.3 \pm 3.6$ |
| 5 | 0.5 | $17.1 \pm 3.0$ | $68.5 \pm 3.4$ |
| 10 | 0.1 | $5.3 \pm 2.5$ | $57.8 \pm 4.3$ |
| 10 | 0.2 | $8.2 \pm 3.3$ | $63.9 \pm 3.6$ |
| 10 | 0.3 | $9.3 \pm 2.0$ | $68.8 \pm 2.6$ |
| 10 | 0.4 | $11.5 \pm 2.7$ | $72.4 \pm 4.0$ |
| 10 | 0.5 | $12.8 \pm 3.9$ | $76.6 \pm 3.7$ |
| 15 | 0.1 | $10.1 \pm 1.1$ | $69.5 \pm 3.3$ |
| 15 | 0.2 | $7.6 \pm 1.2$ | $70.0 \pm 4.6$ |
| 15 | 0.3 | $8.7 \pm 1.5$ | $73.1 \pm 5.1$ |
| 15 | 0.4 | $9.8 \pm 2.0$ | $74.5 \pm 2.5$ |
| 15 | 0.5 | $12.0 \pm 1.8$ | $79.7 \pm 2.3$ |

Table 6: MOLCRAFT Certification Results

| $N$ | $\alpha$ | Error Rate (%) | Certified Fraction (%) |
|---|---|---|---|
| 5 | 0.1 | $7.0 \pm 1.2$ | $40.1 \pm 2.7$ |
| 5 | 0.2 | $11.0 \pm 1.8$ | $47.9 \pm 3.5$ |
| 5 | 0.3 | $12.7 \pm 1.0$ | $55.5 \pm 4.9$ |
| 5 | 0.4 | $15.1 \pm 1.2$ | $63.1 \pm 4.9$ |
| 5 | 0.5 | $16.6 \pm 1.2$ | $68.1 \pm 3.8$ |
| 10 | 0.1 | $5.2 \pm 1.6$ | $58.3 \pm 5.8$ |
| 10 | 0.2 | $6.7 \pm 1.1$ | $64.2 \pm 5.8$ |
| 10 | 0.3 | $7.5 \pm 1.4$ | $67.6 \pm 4.9$ |
| 10 | 0.4 | $9.0 \pm 1.6$ | $71.1 \pm 7.1$ |
| 10 | 0.5 | $9.0 \pm 2.9$ | $73.5 \pm 7.5$ |
| 15 | 0.1 | $7.3 \pm 1.0$ | $55.4 \pm 4.1$ |
| 15 | 0.2 | $9.1 \pm 2.0$ | $59.8 \pm 5.8$ |
| 15 | 0.3 | $9.1 \pm 3.6$ | $68.7 \pm 5.9$ |
| 15 | 0.4 | $9.5 \pm 3.0$ | $69.1 \pm 7.2$ |
| 15 | 0.5 | $11.5 \pm 4.9$ | $76.2 \pm 7.8$ |

Table 7: DECOMPDIFF Certification Results

| $N$ | $\alpha$ | Error Rate (%) | Certified Fraction (%) |
|---|---|---|---|
| 5 | 0.1 | $3.2 \pm 2.2$ | $56.6 \pm 1.4$ |
| 5 | 0.2 | $3.1 \pm 1.9$ | $63.5 \pm 1.2$ |
| 5 | 0.3 | $4.7 \pm 2.0$ | $69.0 \pm 2.4$ |
| 5 | 0.4 | $5.2 \pm 2.4$ | $73.4 \pm 0.8$ |
| 5 | 0.5 | $5.4 \pm 2.6$ | $77.0 \pm 1.2$ |
| 10 | 0.1 | $2.1 \pm 1.6$ | $74.4 \pm 4.0$ |
| 10 | 0.2 | $3.1 \pm 2.5$ | $76.4 \pm 3.5$ |
| 10 | 0.3 | $5.2 \pm 3.1$ | $79.6 \pm 3.4$ |
| 10 | 0.4 | $6.2 \pm 2.8$ | $82.7 \pm 3.6$ |
| 10 | 0.5 | $5.9 \pm 3.0$ | $82.7 \pm 2.1$ |
| 15 | 0.1 | $2.6 \pm 3.2$ | $83.2 \pm 5.7$ |
| 15 | 0.2 | $2.5 \pm 3.2$ | $83.9 \pm 5.0$ |
| 15 | 0.3 | $1.6 \pm 2.7$ | $85.8 \pm 5.3$ |
| 15 | 0.4 | $1.6 \pm 2.8$ | $85.1 \pm 6.2$ |
| 15 | 0.5 | $2.5 \pm 3.1$ | $87.1 \pm 5.2$ |

### E.1.2 DESIGN RESULTS

DRD2_HGRAPH

Table 8: DRD2_HGRAPH Design Results

| $N$ | $\alpha$ | Error Rate (%) | Mean Set Size | Empty Set (%) |
|---|---|---|---|---|
| 3 | 0.1 | $4.2 \pm 0.3$ | $1.9 \pm 0.0$ | $68.8 \pm 1.3$ |
| 3 | 0.2 | $6.6 \pm 0.5$ | $1.8 \pm 0.0$ | $46.1 \pm 1.3$ |
| 3 | 0.3 | $9.1 \pm 0.6$ | $1.8 \pm 0.0$ | $25.3 \pm 1.8$ |
| 3 | 0.4 | $11.0 \pm 0.8$ | $1.6 \pm 0.0$ | $15.0 \pm 1.3$ |
| 3 | 0.5 | $13.0 \pm 0.8$ | $1.5 \pm 0.0$ | $9.9 \pm 0.8$ |
| 5 | 0.1 | $5.4 \pm 1.6$ | $2.4 \pm 0.7$ | $47.9 \pm 2.6$ |
| 5 | 0.2 | $7.9 \pm 2.5$ | $2.1 \pm 0.7$ | $28.7 \pm 2.8$ |
| 5 | 0.3 | $10.4 \pm 3.3$ | $1.8 \pm 0.6$ | $15.8 \pm 2.1$ |
| 5 | 0.4 | $12.4 \pm 4.0$ | $1.6 \pm 0.5$ | $11.3 \pm 1.9$ |
| 5 | 0.5 | $13.9 \pm 4.4$ | $1.4 \pm 0.5$ | $8.8 \pm 1.5$ |
| 7 | 0.1 | $3.6 \pm 0.8$ | $3.1 \pm 0.1$ | $53.0 \pm 1.2$ |
| 7 | 0.2 | $6.3 \pm 0.7$ | $2.6 \pm 0.1$ | $30.0 \pm 1.4$ |
| 7 | 0.3 | $9.0 \pm 1.0$ | $2.1 \pm 0.0$ | $15.8 \pm 0.7$ |
| 7 | 0.4 | $11.2 \pm 1.1$ | $1.8 \pm 0.0$ | $9.2 \pm 0.5$ |
| 7 | 0.5 | $12.9 \pm 1.2$ | $1.7 \pm 0.0$ | $5.9 \pm 0.6$ |

DRD2_SELFEDIT

Table 9: DRD2_SELFEDIT Design Results

| $N$ | $\alpha$ | Error Rate (%) | Mean Set Size | Empty Set (%) |
|---|---|---|---|---|
| 3 | 0.1 | $9.8 \pm 1.3$ | $2.0 \pm 0.0$ | $64.5 \pm 0.5$ |
| 3 | 0.2 | $14.6 \pm 2.2$ | $1.9 \pm 0.1$ | $42.3 \pm 0.2$ |
| 3 | 0.3 | $19.6 \pm 3.2$ | $1.7 \pm 0.0$ | $24.0 \pm 0.9$ |
| 3 | 0.4 | $22.5 \pm 3.2$ | $1.6 \pm 0.0$ | $12.3 \pm 0.1$ |
| 3 | 0.5 | $24.5 \pm 4.0$ | $1.5 \pm 0.0$ | $7.3 \pm 0.6$ |
| 5 | 0.1 | $9.5 \pm 1.2$ | $2.6 \pm 0.1$ | $53.0 \pm 1.5$ |
| 5 | 0.2 | $14.0 \pm 2.4$ | $2.3 \pm 0.1$ | $33.3 \pm 2.6$ |
| 5 | 0.3 | $20.7 \pm 2.1$ | $2.0 \pm 0.0$ | $14.9 \pm 0.6$ |
| 5 | 0.4 | $25.8 \pm 2.6$ | $1.7 \pm 0.1$ | $8.1 \pm 1.1$ |
| 5 | 0.5 | $28.8 \pm 3.1$ | $1.6 \pm 0.0$ | $4.2 \pm 0.4$ |
| 7 | 0.1 | $11.8 \pm 1.3$ | $3.1 \pm 0.1$ | $46.6 \pm 0.8$ |
| 7 | 0.2 | $19.5 \pm 3.3$ | $2.6 \pm 0.0$ | $22.9 \pm 3.5$ |
| 7 | 0.3 | $24.8 \pm 6.3$ | $2.1 \pm 0.1$ | $12.2 \pm 1.9$ |
| 7 | 0.4 | $28.6 \pm 7.1$ | $1.8 \pm 0.1$ | $7.4 \pm 1.7$ |
| 7 | 0.5 | $31.6 \pm 6.3$ | $1.7 \pm 0.0$ | $4.0 \pm 0.6$ |

QED_HGRAPH

Table 10: QED_HGRAPH Design Results

| $N$ | $\alpha$ | Error Rate (%) | Mean Set Size | Empty Set (%) |
|---|---|---|---|---|
| 3 | 0.1 | $10.1 \pm 0.7$ | $1.2 \pm 0.0$ | $70.7 \pm 1.9$ |
| 3 | 0.2 | $17.0 \pm 0.6$ | $1.3 \pm 0.0$ | $45.4 \pm 1.6$ |
| 3 | 0.3 | $21.2 \pm 1.4$ | $1.3 \pm 0.1$ | $26.8 \pm 0.9$ |
| 3 | 0.4 | $25.2 \pm 1.5$ | $1.3 \pm 0.0$ | $13.2 \pm 1.2$ |
| 3 | 0.5 | $28.2 \pm 1.7$ | $1.2 \pm 0.0$ | $6.9 \pm 0.6$ |
| 5 | 0.1 | $12.2 \pm 0.7$ | $2.6 \pm 0.0$ | $27.2 \pm 0.2$ |
| 5 | 0.2 | $18.8 \pm 1.1$ | $2.3 \pm 0.1$ | $7.1 \pm 0.2$ |
| 5 | 0.3 | $24.4 \pm 1.2$ | $1.8 \pm 0.0$ | $0.0 \pm 0.0$ |
| 5 | 0.4 | $27.7 \pm 1.3$ | $1.6 \pm 0.1$ | $0.0 \pm 0.0$ |
| 5 | 0.5 | $29.9 \pm 1.0$ | $1.5 \pm 0.1$ | $0.0 \pm 0.0$ |
| 7 | 0.1 | $10.4 \pm 1.2$ | $2.5 \pm 0.1$ | $56.6 \pm 0.8$ |
| 7 | 0.2 | $17.5 \pm 1.2$ | $2.0 \pm 0.0$ | $31.7 \pm 1.8$ |
| 7 | 0.3 | $22.0 \pm 1.4$ | $1.7 \pm 0.1$ | $18.9 \pm 1.6$ |
| 7 | 0.4 | $26.1 \pm 1.5$ | $1.4 \pm 0.0$ | $9.4 \pm 1.5$ |
| 7 | 0.5 | $29.1 \pm 1.6$ | $1.3 \pm 0.0$ | $4.9 \pm 1.1$ |

QED_SELFEDIT

Table 11: QED_SELFEDIT Design Results

| $N$ | $\alpha$ | Error Rate (%) | Mean Set Size | Empty Set (%) |
|---|---|---|---|---|
| 3 | 0.1 | $10.1 \pm 0.4$ | $1.2 \pm 0.0$ | $64.8 \pm 1.8$ |
| 3 | 0.2 | $14.4 \pm 0.9$ | $1.3 \pm 0.1$ | $45.2 \pm 1.5$ |
| 3 | 0.3 | $18.8 \pm 1.4$ | $1.3 \pm 0.1$ | $27.1 \pm 1.8$ |
| 3 | 0.4 | $21.5 \pm 1.9$ | $1.2 \pm 0.1$ | $13.3 \pm 1.0$ |
| 3 | 0.5 | $23.4 \pm 1.9$ | $1.2 \pm 0.0$ | $7.0 \pm 1.0$ |
| 5 | 0.1 | $11.9 \pm 0.8$ | $2.7 \pm 0.1$ | $26.0 \pm 2.4$ |
| 5 | 0.2 | $20.1 \pm 1.2$ | $2.3 \pm 0.1$ | $7.0 \pm 0.7$ |
| 5 | 0.3 | $25.8 \pm 1.4$ | $1.9 \pm 0.1$ | $0.0 \pm 0.0$ |
| 5 | 0.4 | $28.4 \pm 1.5$ | $1.6 \pm 0.1$ | $0.0 \pm 0.0$ |
| 5 | 0.5 | $29.9 \pm 2.1$ | $1.5 \pm 0.2$ | $0.0 \pm 0.0$ |
| 7 | 0.1 | $18.2 \pm 2.7$ | $2.1 \pm 0.2$ | $56.7 \pm 1.5$ |
| 7 | 0.2 | $26.9 \pm 3.8$ | $1.9 \pm 0.2$ | $34.7 \pm 2.9$ |
| 7 | 0.3 | $34.5 \pm 3.0$ | $1.6 \pm 0.1$ | $18.3 \pm 2.2$ |
| 7 | 0.4 | $39.2 \pm 3.0$ | $1.4 \pm 0.1$ | $10.2 \pm 2.5$ |
| 7 | 0.5 | $42.1 \pm 2.5$ | $1.3 \pm 0.0$ | $5.7 \pm 1.6$ |

TARGETDIFF

Table 12: TARGETDIFF Design Results

| $N$ | $\alpha$ | Error Rate (%) | Mean Set Size | Empty Set (%) |
|---|---|---|---|---|
| 5 | 0.1 | $10.0 \pm 1.5$ | $2.5 \pm 0.1$ | $37.9 \pm 2.6$ |
| 5 | 0.2 | $17.4 \pm 1.5$ | $2.2 \pm 0.1$ | $21.4 \pm 3.1$ |
| 5 | 0.3 | $27.2 \pm 3.5$ | $2.0 \pm 0.1$ | $8.6 \pm 0.0$ |
| 5 | 0.4 | $34.2 \pm 2.9$ | $1.7 \pm 0.1$ | $2.1 \pm 1.4$ |
| 5 | 0.5 | $37.0 \pm 1.2$ | $1.6 \pm 0.0$ | $0.4 \pm 0.7$ |
| 10 | 0.1 | $9.4 \pm 2.7$ | $3.7 \pm 0.1$ | $21.9 \pm 8.0$ |
| 10 | 0.2 | $22.5 \pm 3.5$ | $2.9 \pm 0.0$ | $6.2 \pm 3.5$ |
| 10 | 0.3 | $29.4 \pm 0.9$ | $2.3 \pm 0.0$ | $1.2 \pm 0.0$ |
| 10 | 0.4 | $35.6 \pm 0.9$ | $1.8 \pm 0.0$ | $0.0 \pm 0.0$ |
| 10 | 0.5 | $37.5 \pm 3.5$ | $1.6 \pm 0.1$ | $0.0 \pm 0.0$ |
| 15 | 0.1 | $11.2 \pm 1.8$ | $4.7 \pm 0.1$ | $15.0 \pm 5.3$ |
| 15 | 0.2 | $21.9 \pm 2.7$ | $3.5 \pm 0.0$ | $3.8 \pm 1.8$ |
| 15 | 0.3 | $31.9 \pm 0.9$ | $2.4 \pm 0.0$ | $1.2 \pm 1.8$ |
| 15 | 0.4 | $36.9 \pm 2.7$ | $1.8 \pm 0.0$ | $0.6 \pm 0.9$ |
| 15 | 0.5 | $38.1 \pm 2.7$ | $1.6 \pm 0.1$ | $0.0 \pm 0.0$ |

MOLCRAFT

Table 13: MOLCRAFT Design Results

| $N$ | $\alpha$ | Error Rate (%) | Mean Set Size | Empty Set (%) |
|---|---|---|---|---|
| 5 | 0.1 | $9.8 \pm 1.2$ | $2.6 \pm 0.1$ | $40.7 \pm 1.9$ |
| 5 | 0.2 | $18.7 \pm 1.4$ | $2.3 \pm 0.2$ | $20.7 \pm 5.6$ |
| 5 | 0.3 | $26.0 \pm 3.1$ | $2.0 \pm 0.1$ | $7.7 \pm 0.7$ |
| 5 | 0.4 | $29.3 \pm 5.3$ | $1.7 \pm 0.0$ | $2.4 \pm 0.0$ |
| 5 | 0.5 | $32.9 \pm 4.2$ | $1.6 \pm 0.0$ | $1.2 \pm 1.2$ |
| 10 | 0.1 | $9.9 \pm 1.3$ | $3.6 \pm 0.3$ | $22.2 \pm 3.3$ |
| 10 | 0.2 | $18.2 \pm 2.0$ | $3.0 \pm 0.2$ | $9.9 \pm 2.1$ |
| 10 | 0.3 | $25.1 \pm 0.7$ | $2.2 \pm 0.1$ | $2.9 \pm 1.4$ |
| 10 | 0.4 | $28.0 \pm 1.4$ | $1.8 \pm 0.1$ | $2.1 \pm 0.7$ |
| 10 | 0.5 | $29.6 \pm 2.5$ | $1.7 \pm 0.1$ | $0.4 \pm 0.7$ |
| 15 | 0.1 | $11.1 \pm 0.0$ | $4.3 \pm 0.0$ | $16.0 \pm 0.0$ |
| 15 | 0.2 | $19.8 \pm 0.0$ | $3.2 \pm 0.0$ | $1.2 \pm 0.0$ |
| 15 | 0.3 | $23.5 \pm 0.0$ | $2.2 \pm 0.0$ | $1.2 \pm 0.0$ |
| 15 | 0.4 | $27.2 \pm 0.0$ | $1.8 \pm 0.0$ | $1.2 \pm 0.0$ |
| 15 | 0.5 | $28.4 \pm 0.0$ | $1.6 \pm 0.0$ | $1.2 \pm 0.0$ |

DECOMPDIFF

Table 14: DECOMPDIFF Design Results

| $N$ | $\alpha$ | Error Rate (%) | Mean Set Size | Empty Set (%) |
|---|---|---|---|---|
| 5 | 0.1 | $9.6 \pm 2.6$ | $2.6 \pm 0.3$ | $25.0 \pm 2.2$ |
| 5 | 0.2 | $16.7 \pm 4.7$ | $2.3 \pm 0.2$ | $11.7 \pm 0.7$ |
| 5 | 0.3 | $24.2 \pm 0.7$ | $2.0 \pm 0.0$ | $2.5 \pm 2.2$ |
| 5 | 0.4 | $26.7 \pm 0.7$ | $1.7 \pm 0.1$ | $1.2 \pm 0.0$ |
| 5 | 0.5 | $28.8 \pm 1.2$ | $1.6 \pm 0.1$ | $0.8 \pm 0.7$ |
| 10 | 0.1 | $11.2 \pm 0.5$ | $3.4 \pm 0.1$ | $13.5 \pm 0.9$ |
| 10 | 0.2 | $19.2 \pm 0.1$ | $2.7 \pm 0.0$ | $2.6 \pm 0.0$ |
| 10 | 0.3 | $24.4 \pm 3.6$ | $2.1 \pm 0.1$ | $0.0 \pm 0.0$ |
| 10 | 0.4 | $26.3 \pm 0.9$ | $1.7 \pm 0.1$ | $0.0 \pm 0.0$ |
| 10 | 0.5 | $27.6 \pm 0.9$ | $1.6 \pm 0.1$ | $0.0 \pm 0.0$ |
| 15 | 0.1 | $9.6 \pm 0.9$ | $4.3 \pm 0.4$ | $8.3 \pm 2.7$ |
| 15 | 0.2 | $19.2 \pm 3.6$ | $2.9 \pm 0.1$ | $1.3 \pm 1.8$ |
| 15 | 0.3 | $23.7 \pm 4.5$ | $2.3 \pm 0.1$ | $0.0 \pm 0.0$ |
| 15 | 0.4 | $26.9 \pm 3.6$ | $1.8 \pm 0.0$ | $0.0 \pm 0.0$ |
| 15 | 0.5 | $27.6 \pm 2.7$ | $1.7 \pm 0.1$ | $0.0 \pm 0.0$ |

## E.2 COMPARING DIFFERENT TEST STATISTICS

In this section, we include additional results for Section 4.5 comparing different test statistics. While the power varies based on the choice of the test statistic, CONFHIT attains valid coverage across statistics.

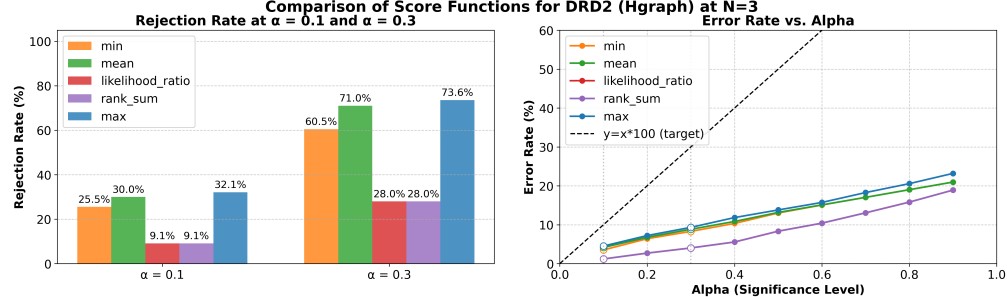

Figure 7: Comparison of score statistics on DRD2 (HGRAPH2GRAPH, N=3). Left: rejection (power) at $\alpha = 0.1, 0.3$. Right: error vs. $\alpha$

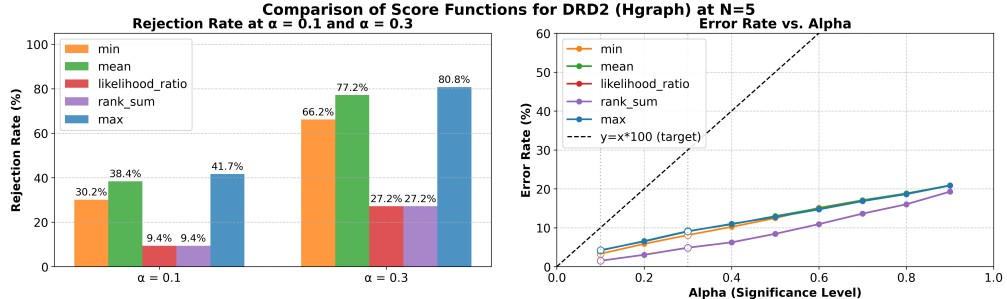

Figure 8: Comparison of score statistics on DRD2 (HGRAPH2GRAPH, N=5). Left: rejection (power) at $\alpha = 0.1, 0.3$. Right: error vs. $\alpha$

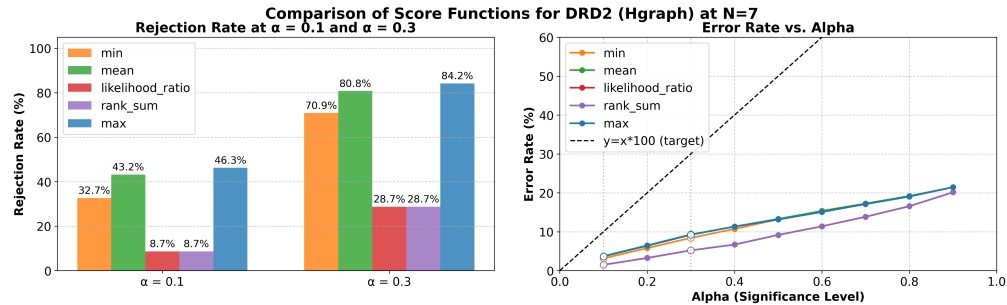

Figure 9: Comparison of score statistics on DRD2 (HGRAPH2GRAPH, N=7). Left: rejection (power) at $\alpha = 0.1, 0.3$. Right: error vs. $\alpha$

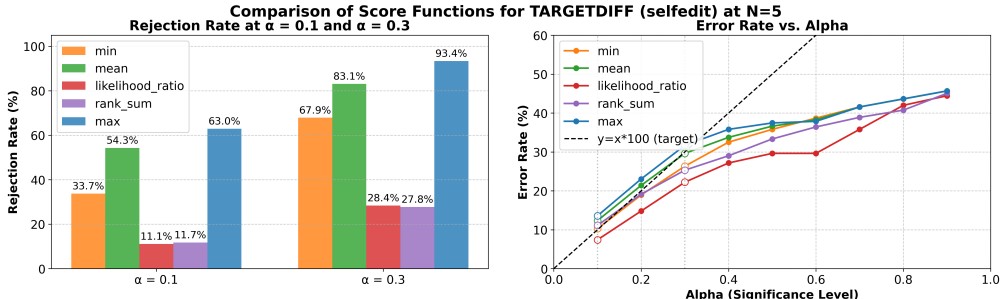

Figure 10: Comparison of score statistics on SBDD (TARGETDIFF, N=5). Left: rejection (power) at $\alpha = 0.1, 0.3$. Right: error vs. $\alpha$

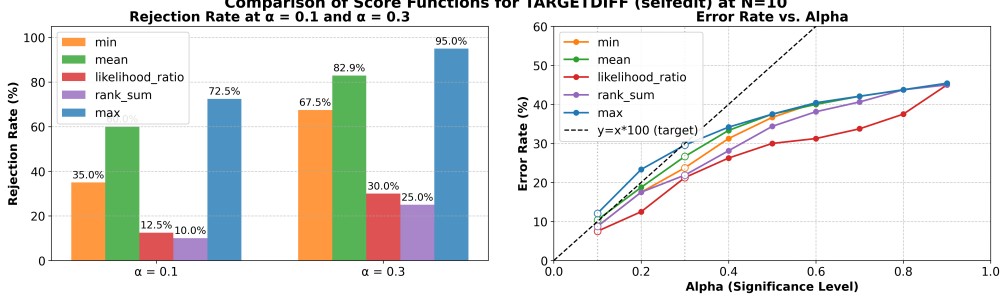

Figure 11: Comparison of score statistics on SBDD (TARGETDIFF, N=10). Left: rejection (power) at $\alpha = 0.1, 0.3$. Right: error vs. $\alpha$

### E.3 P-VALUES

In this section, we show the validity of conformal p-value plots described in section 4.5, across tasks and datasets. In each panel, the distribution remains close to the uniform (dashed line; see also the reported KL divergence from uniformity), confirming approximate validity of our p-values (albeit slightly conservative).

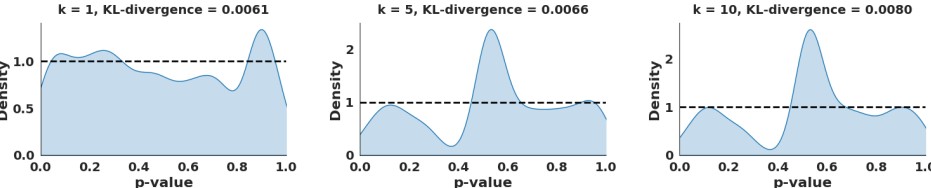

Figure 12: Density of conformal nested p-values for QED using SELFEDIT at set sizes $k = 3, 5, 7$.

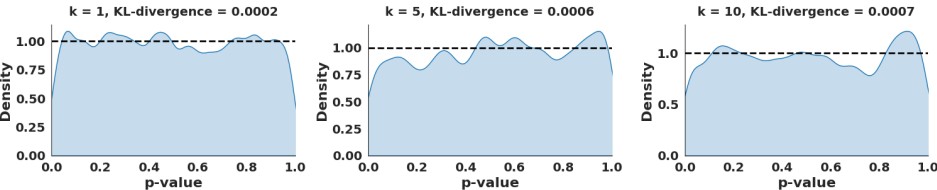

Figure 13: Density of conformal nested p-values for DRD2 using HGRAPH at set sizes $k = 3, 5, 7$.

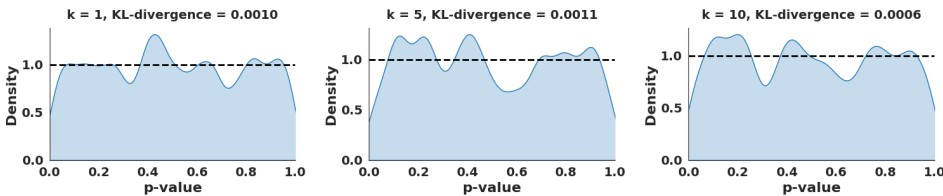

Figure 14: Density of conformal nested p-values for SBDD using TARGETDIFF at set sizes $k = 1, 5, 10$.

## F EFFECT OF PREDICTOR QUALITY

In this section, we investigate how the accuracy of the predictor affects the performance of CONFHIT. We consider three scenarios:

1. Normal predictor where the calibrated classifier outputs $p = \Pr(\text{active} \mid x)$.
2. Noisy predictor obtained by adding uniform Gaussian noise, giving

$$\tilde{p} = \frac{p + \mathcal{N}(0, 1)}{2}.$$

3. Inverse predictor obtained by replacing $p$ with $1 - p$.

Figure 15 illustrates how CONFHIT behaves under these perturbation settings. Even when the predictor is corrupted or inverted, the method maintains valid error control, with empirical error rates remaining below the target line $y = 100\alpha$. This demonstrates that the statistical validity of CONFHIT does not rely on predictor accuracy.

Predictor quality affects only the power. As the predictor becomes less informative, the conformal sets become less efficient, which leads to a higher frequency of empty sets or larger sets depending on the task. Despite this expected degradation, the method consistently preserves its coverage guarantees.

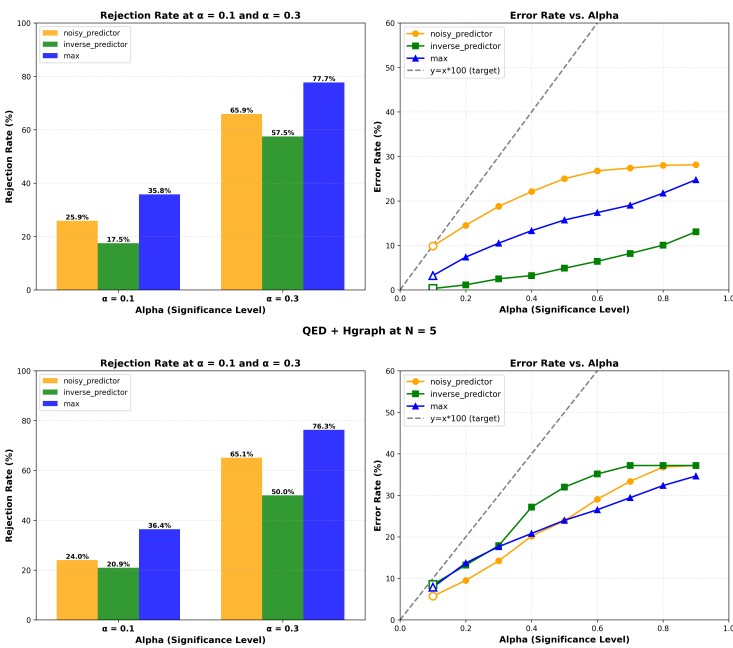

Figure 15: Effect of predictor quality on CONFHIT's error rate on the CMO task. In both DRD2 (top) and QED (bottom) using the Hgraph model, noisy and inverse predictors exhibit reduced power but maintain valid error control across all values of $\alpha$.

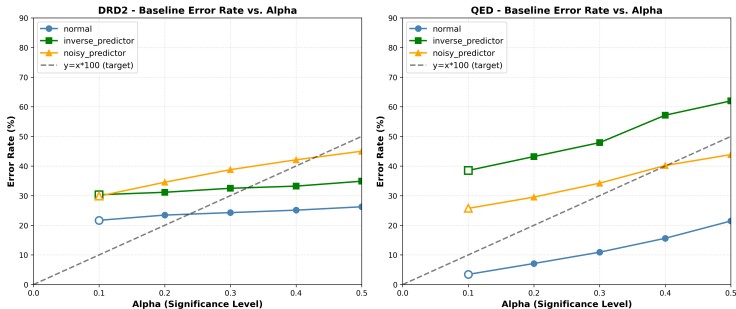

Figure 16: Results for heuristic baseline with predicted probabilities (while varying predictor quality), showing rejection rates and error rates for QED with $N = 5$ using the Hgraph model. As predictor quality deteriorates, the performance quickly drops with severe violation of the error control.

# G   ADDITIONAL BASELINES

## G.1   HEURISTIC BASELINE

We evaluate a simple confidence based heuristic. Given a predicted probability $\hat{p}$ of success, the heuristic selects the smallest $n$ such that $(1 - \hat{p})^n \leq \alpha$. This provides a natural baseline that depends only on classifier scores without any uncertainty calibration.

The results in Figures 16 and show that this heuristic does not reliably control the error rate. When the predictor is accurate, the error rates sometimes follow the target line, but under noisy or inverted predictors the error increases sharply and often surpasses the desired level. The degradation is visible across both DRD2 and QED: the error curves drift upward as the predictor worsens, and the rejection rates fluctuate significantly, indicating unstable behavior.

In contrast, ConfHit remains below the target line in all cases. The predictor quality affects only the power, whereas the heuristic baseline loses error control as soon as the predictor becomes unreliable.

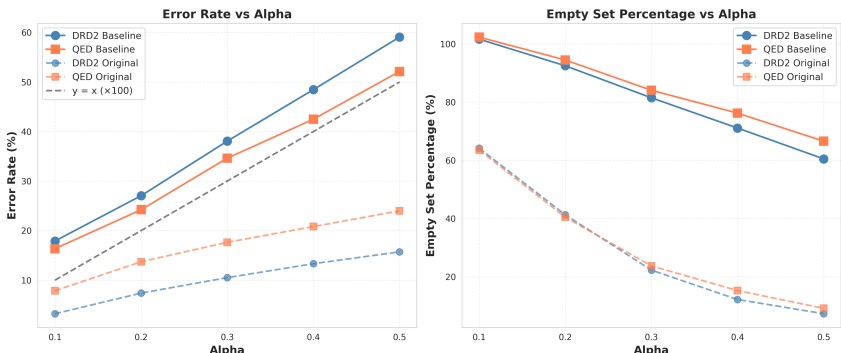

Figure 17: Conformal LM baseline for molecule optimization. Error rate (left) and empty–set percentage (right) versus $\alpha$ for QED and DRD2 at N = 5 using the Hgraph model. The method breaks coverage and underperforms CONFHIT despite relying on an oracle that is unavailable in real-world settings.

The empirical curves highlight that the heuristic transitions from reasonable to poor performance, while ConfHit maintains stable guarantees across all predictor settings.

### G.2 COMPARISON TO ORACLE BASED METHODS

We adapt an oracle-based CLM (Quach et al., 2023) baseline as depicted in Algorithm 3 to use the *sum* of prediction scores as the total confidence of a candidate set, without performing any density correction or filtering of low–quality samples. As shown in Figure 17, this baseline fails to maintain valid guarantees: the empirical error rate systematically exceeds the target level, and the empty–set percentages deviate sharply from expected behavior across all $\alpha$. Despite relying on an idealized oracle, this approach also exhibits *lower power* than CONFHIT, whereas CONFHIT attains both higher detection power *and* valid coverage due to its calibrated scoring rule and density-aware correction. Finally, we emphasize that this baseline is not directly comparable to CONFHIT, since it assumes access to a perfect oracle predictor—an unrealistic requirement in practical molecular design workflows.

---

**Algorithm 3** Conformal Sampling for Constrained Molecular Optimization

---

**Input:** Input molecule $x$; generator $p_\theta(\cdot \mid x)$; scoring function $\mathcal{F}$; calibrated threshold $\lambda$; sampling budget $k_{\max}$
**Output:** Candidate set $C_\lambda$ certified to contain at least one improved molecule
1: $C_\lambda \leftarrow \varnothing$ {Initialize output set}
2: **for** $k = 1$ **to** $k_{\max}$ **do**
3:      Sample $y_k \sim p_\theta(\cdot \mid x)$ {Generate candidate}
4:      $C_\lambda \leftarrow C_\lambda \cup \{y_k\}$ {Add to set}
5:      **if** $\mathcal{F}(x, C_\lambda) \geq \lambda$ **then**
6:          **break** {Set certified}
7:      **end if**
8: **end for**
9: **return** $C_\lambda$

---

## H ROBUSTNESS AND DIAGNOSTICS FOR DENSITY RATIO ESTIMATION

### H.1 SENSITIVITY ANALYSIS

We study the effect of misspecifying the likelihood ratios by applying power transformations $w^\gamma$ across a range of exponents $\gamma$. Figure 18 shows how such transformations affect the KL divergence of the p-values of the negative samples, together with the resulting error rate and rejection (empty set) rate. In practice, such results can be used to judge whether the analysis is robust enough. Here, we

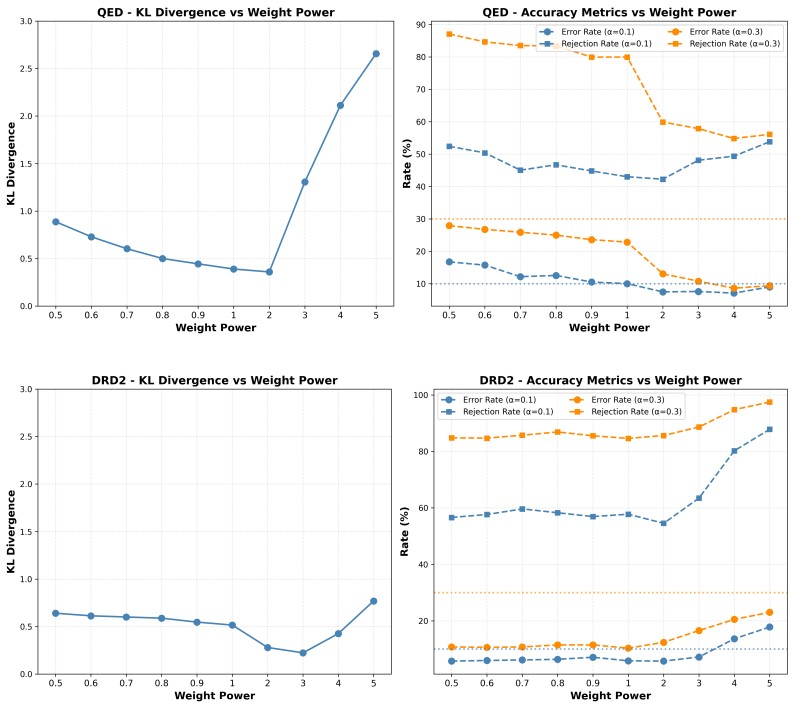

Figure 18: Effect of power transformations $w^\gamma$ of likelihood-ratio weights on QED and DRD2 using the HGraph2Graph model at $N = 5$ at different $\alpha$. Across all transformations, CONFHIT maintains a graceful degradation of error control with major changes in terms of the rejection rate.

additionally evaluate the error control performance under such perturbation to also show the robust performance of CONFHIT.

The KL divergence here measures the deviation of the empirical p-value distribution from the Uniform(0,1) distribution. Smaller values indicate that the transformed likelihood ratios still produce p-values close to uniform, while larger values reflect increasing distortion away from the ideal uniform shape. In both the datasets, we see that the KL divergence is only moderately affected under moderate misspecification. Only on amplifying the weights ($\gamma > 3$) causes the KL divergence to rise sharply in the QED case we notice substantial deviation from uniformity.

The error rate consistently stays near the target level across all transformations for $\alpha = 0.1$ and below the target level for $\alpha = 0.3$. Only the rejection rate is affected, with lower rejection occurring when the KL divergence is small and higher rejection when the p-values deviate more strongly from uniform. This shows that mild misspecification primarily adjusts the conservativeness of the method without affecting its validity.

A similar pattern appears for DRD2 . Flattened weights reduce the KL divergence, while large values of $\gamma$ inflate it. Error control remains stable throughout, whereas the rejection rate increases in the high-KL regime corresponding to more severe distortions of the p-value distribution. Overall, these results demonstrate that ConfHit is robust to moderate incorrect specification of the likelihood ratios.

## H.2 SCAFFOLD SPLITTING ON CALIBRATION SET

To assess the robustness of the calibration procedure, we perform a scaffold split on the calibration negatives. We select the top 30 scaffolds that also appear in the test set, producing a nontrivial distribution shift that reflects realistic drug discovery scenarios where scaffold diversity plays an important role. This setting provides a diagnostic test: if the calibration procedure is effective, the resulting p-values on this shifted split should remain close to uniform.

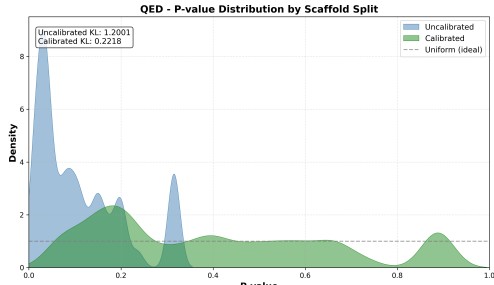 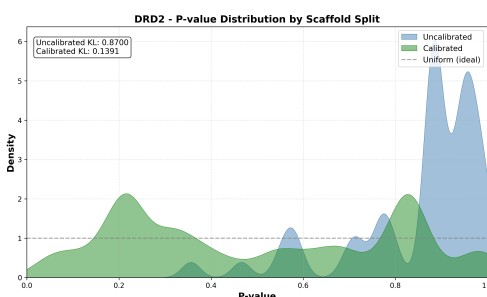

(a) QED + Hgraph: Calibration significantly reduces deviation of p-value distribution from uniformity.

(b) DRD2 + Hgraph: Calibrated p-values closely match the uniform distribution.

Figure 19: P-value distributions on scaffold-split calibration negatives for QED (left) and DRD2 (right). In both cases, calibration produces p-values that closely follow the uniform distribution, indicating that the density correction remains reliable under scaffold-based distribution shift.

Figures 19a and 19b show the p-value distributions before and after applying the density ratio correction. In both QED and DRD2, the uncalibrated p-values deviate substantially from the ideal uniform distribution, as indicated by large KL divergences. After calibration, the p-values become significantly closer to uniform, with much lower KL divergences and a visibly more homogeneous density across the interval.

The close match between the calibrated p-values and the uniform reference suggests that the calibration procedure successfully corrects for distributional differences tied to scaffold variation. This indicates that the method is likely to remain well calibrated on the test set, even when the chemical space differs from the training distribution. In practical applications, having partial knowledge of the test set's chemical space enables a simple sanity check: if the p-values on the corresponding scaffolds appear close to uniform, one can be confident that the method will behave reliably on the actual test compounds.

## H.3 BALANCE CHECKS

Finally, To diagnose the effect of density correction, we compare the feature means of the calibration set with the test set before and after reweighting as discussed in Section 3.3. Figures 20 and 21 show the mean and standard deviation of each feature dimension, where post-calibration test features are weighted by the likelihood ratio $p_{\text{cal}}(x)/p_{\text{test}}(x)$.

For QED (Figure 20), the pre-calibration test features differ substantially from the calibration distribution, especially in the first dimension. After applying density correction, the weighted test features align much more closely with the calibration means across all dimensions. The cosine distance between the mean feature vectors decreases from $0.0623$ to $0.0105$, indicating a strong improvement in distributional matching.

A similar trend appears for DRD2 (Figure 21), where large discrepancies in several dimensions are substantially reduced after reweighting. The cosine distance drops from $0.0491$ to $0.0075$, confirming that the corrected test distribution becomes much closer to the calibration distribution.

These balance checks provide a practical diagnostic: given an unlabeled test set, one can evaluate whether density correction brings its feature distribution closer to the calibration set, indicating that the calibration procedure is likely to remain valid.

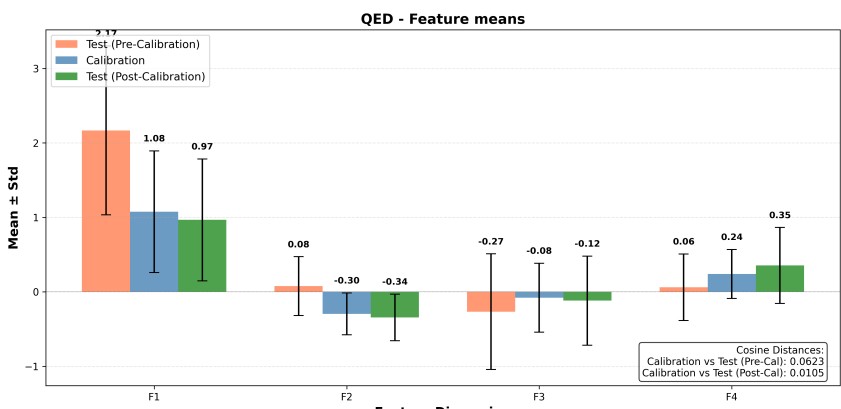

Figure 20: Feature means for QED using the Hgraph model at $N = 5$. Density correction aligns the test features more closely with the calibration distribution.

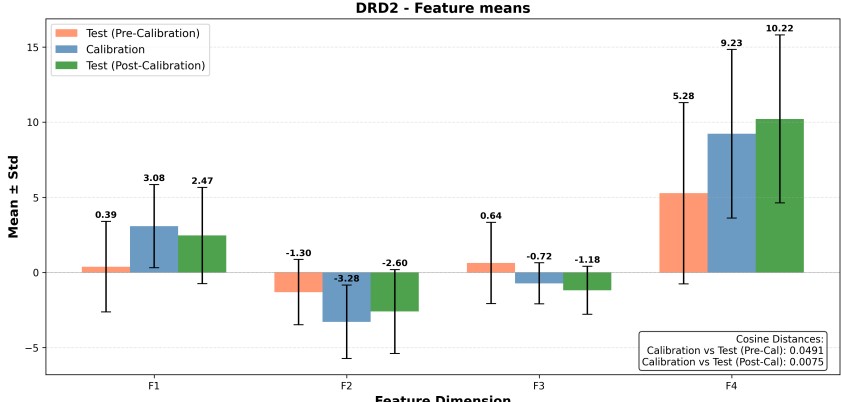

Figure 21: Feature means for DRD2 using the Hgraph model at $N = 5$. Density correction reduces the discrepancy between calibration and test distributions.

