# OpenReview forum: "ConfHit: Conformal Generative Design with Oracle-Free Guarantees"
_ICLR.cc/2026/Conference — ICLR 2026 Poster_

### Official Review · Reviewer_ZnoF · 2025-10-21

**Soundness:** 2
**Presentation:** 4
**Contribution:** 4
**Rating:** 6
**Confidence:** 4

**Summary:**

The authors present ConfHit, a conformal prediction technique that can be used for two applications:

(i) Certification: Given a set of examples, assert whether at least one example fullfills some property (called *hit*), with the guarantee that incorrectly detecting a hit comes at probability $\leq \alpha$.

(ii) Design: Generate a minimal set of examples using a generative model, such that there exists at least one hit, with probability $\geq 1- \alpha$ (built on top of the method for (i)).

Importantly, and in contrast to prior work, ConfHit does not assume access to an oracle for performing calibration, but only assumes access to a set of examples without a hit.

The method is evaluated on constrained molecule optimisation and structure-based drug discovery and performs favorably in comparison to naive baselines.

**Strengths:**

Overall, I believe that this paper is strong and should be accepted.

1. The writing is phenomenal. Everything is clear, the notation is wisely chosen. Also, Figure 1 is excellent.

2. The motivation is well-justified. The authors tackle a relevant limitation of existing approaches.

3. Rigorous theoretical analysis is present.

4. The method is relatively simple.

5. Strong experimental evaluation.

**Weaknesses:**

A great concern I have is the fact that ConfHit crucially hinges on a perfect estimate of the density ratio $w$. While Theorem 3.4 nicely quantifies the effect of estimation error of $w$ on the guarantee, the corrected bound is obviously not computable in practice. The authors motivate their technique via the need for guarantees for erroneous generative models. However, the technique itself relies on a erroneous density estimate, for which the guarantee does no longer hold. Thus, it seems that ConfHit shifts the problem from one ML model to another ML model, but it does not actually solve the problem. The authors write that they use kernel density estimation on top of a neural network representation to estimate $w$. Is there anything that suggests that trusting this density ratio estimate is more reasonable than simply trusting the outputs of the generative model? Density estimation is a harder problem than generative modeling (especially in high dimensions), so I have doubts.

The authors briefly discuss this point in their limitations paragraph of the conclusion, but I think it would be important to stress this crucial issue more (ideally, even in the manuscript's abstract) and go into why estimating $w$ should be more reliable than estimating the generative model. I am willing to increase my score if the authors convincingly address this point.

**Questions:**

* l.10: *"Recent advances extend CP to provide guarantees in language generation (Angelopoulos et al., 2021; Quach et al., 2023)"* I believe that [1, 2] should also be cited here.

* l.169: That means we discard all data that fulfills the property of interest? I wonder whether the statistical efficiency of the method could be improved by incorporating these examples somehow.

* Algorithm 1: Maybe I missed it, but how large should we choose the "original" set $\lbrace X_{n+j} \rbrace_{j=1}^N$? Obviously, if $N$ is too small, the set $\lbrace k \in [N]: p_k \leq \alpha \rbrace$ will be empty. What do we do in this case? Reject returning a set? I believe that this corner case is missing in the algorithm.

[1] Kladny, Klaus-Rudolf, Bernhard Schölkopf, and Michael Muehlebach. "Conformal Generative Modeling with Improved Sample Efficiency through Sequential Greedy Filtering." International Conference on Learning Representations (2025).

[2] Shahrokhi, Hooman, et al. "Conformal prediction sets for deep generative models via reduction to conformal regression." Uncertainty in Artificial Intelligence (2025).

---

> ### Author Response · Authors · 2025-11-21
> **Author response 1**
>
> We thank the reviewer for their thoughtful, detailed, and generous comments. We are delighted that you found the paper clearly written, theoretically rigorous, and well-motivated, and we are deeply grateful for your strong overall endorsement. Your insightful feedback on the reliability of the density ratio estimation was especially valuable to us. We have carefully addressed this concern and further strengthened the manuscript accordingly.
>
> 1. **Reliability of density ratio estimate**
>    We fully agree that the validity of ConfHit hinges on a reliable estimate of the density ratio w(x), and we appreciate your observation that this is a crucial issue deserving deeper discussion. In response, we have expanded our analyses to clarify why the density-ratio step is not merely “shifting” the problem to another model but rather *providing new capabilities to generative tasks*, and ConfHit is indeed more robust than the generative model itself as demonstrated in our new experiments.
>     1. **ConfHit provides new capabilities instead of shifting the problem.** Generative models produce complex samples without any built-in notion of reliability; estimating w(x)  instead isolates the specific source of uncertainty (distributional mismatch) into a single scalar quantity that can be *directly validated and stress-tested* (see our points d-e). In other words, ConfHit does not transfer the problem; it makes the uncertainty measurable and provide new capabilities to generative tasks for practical implementation. Also, the difference between historical assay data and newly generated molecules is not unique to our approach: it is a long-standing issue in this field and in nearly all conformal prediction-related works \[1,2,3\].
>     2. **Density ratio estimation may not be as hard.**
>         1. **Statistical advantage.** Estimating the density ratio is statistically simpler than estimating the density directly, as it only requires *relative likelihoods* rather than absolute probabilities. This distinction is well established in the literature on covariate shift correction and uncertainty quantification (e.g., [4,5]). We highlight this explicitly in the newly added Section 3.3 on robustness to estimated weights.
>
>         2. **Practical estimation**: As density ratio estimation is widely used in importance adjustment, covariate shift, and causal inference, there is a rich toolbox we can borrow from these mature literatures. In practice, the ratio can be learned using kernel or classification-based methods that are empirically robust. We have clarified our implementation, noting that we use kernel density estimation following the success of prior work (CoDrug), and we recognize that alternative estimators (e.g., logistic regression-based likelihood ratio estimation) can also be employed effectively.
>     3. **ConfHit’s robustness & new guidance on diagnosis**. We note several newly added results to show the robustness of ConfHit to estimation error, including (i) theory that bounds the degradation, (ii) empirical robustness, (iii) three new diagnosis methods. A more detailed analysis of of these approaches is in response to Reviewer VPnq
>        1. **Bound on degradation**. The theoretical bound (Theorem X) ensures that coverage degradation under misspecified w(x) is controlled by the magnitude of the estimation error.
>        2. **Empirical robustness**. Our empirical error control testifies the robustness of ConfHit to estimated density ratios. We include new sensitivity experiments (Appendix H.1) where we perturb the estimated w(x). ConfHit maintains near-nominal coverage even with moderate mis-specification.
>        3. **Three new diagnosis methods**. We introduce simple diagnostics practitioners can run before certification, including (1) a *balance check* comparing calibration and test statistics, (2) validation shift approach, and (3) a *sensitivity probe* to examine the worst-case results under certain perturbations in w(x). These approaches are discussed in the new Section 3.3, with new experiments in Appendix H and discussion on the results in Section 4.5.

---

> > ### Author Response · Authors · 2025-11-21
> > **Author response 2**
> >
> > 1. **Reliability of density ratio estimate** (cont)
> >     4. **ConfHit is more robust than the generative model alone.** We note our newly added experiments in Appendix G.1 on the error control of ConfHit and a naive baseline which only uses model outputs. The baseline method uses predicted probability of success to select the minimal sample size that contains at least a hit with probability $1-\\alpha$. This baseline quickly incurs high error rates as the predictor quality deteriorates; in contrast, ConfHit remains valid (as it is model agnostic) with estimated weights. We believe this demonstrates that principled uncertainty quantification is indeed more robust than trusting the model itself.
> >     5. Finally, besides the solutions above, in practice, one could bring the calibration distribution closer to the test distribution by obtaining labels for some of the generated samples, and improve the coverage of the calibration data for the new samples.
> > 2. **Missing references**
> >    Thank you for pointing out the relevant literature\! We have cited Kladny et al. when motivating resource-limited scenarios, and have properly referred to both in the updated manuscript.
> > 3. **Use of positive samples**
> >
> > This is an insightful point. We clarify that we do not discard positive examples from the training set: they are used in training the feature extraction/scoring model. Because the power of ConfHit depends directly on the accuracy and discrimination ability of this predictor, the positive samples already contribute to statistical efficiency by improving the learned representation and the score. However, exploring whether they can be leveraged even more directly within the conformal calibration step is an interesting direction for future work.
> >
> > 4. **Choice of “original data” size**
> >    The calibration set size is chosen based on available data, and the validity of ConfHit is finite-sample and does not rely on the sample size. In our experiments, the calibration dataset sizes are in the order of 10,000 data points(for SBDD task obtained from Cross-docked) to 20,000 points (for the CMO task).
> > 5. **Corner cases with small N**
> >    Thank you for pointing out this important corner case\! Yes, when N is too small so that the confidence in a hit is not enough, we would fail to reject the null for any batch, and signal “failure to certify”. Notably, this is a feature instead of the failure of ConfHit: we make this output explicit so that practitioners know how to interpret the results. We have added this case in Algorithm 1 with a brief discussion at the end of Section 3.2.
> >
> >
> >
> > \[1\] Krstajic, Damjan. "Critical assessment of conformal prediction methods applied in binary classification settings." Journal of Chemical Information and Modeling 61.10 (2021): 4823-4826.
> > \[2\] Fannjiang, Clara, and Ji Won Park. "Reliable algorithm selection for machine learning-guided design." arXiv preprint arXiv:2503.20767 (2025).
> > \[3\] Laghuvarapu, Siddhartha, Zhen Lin, and Jimeng Sun. "Codrug: Conformal drug property prediction with density estimation under covariate shift." Advances in Neural Information Processing Systems 36 (2023): 37728-37747.
> > \[4\] Sugiyama, Masashi, Taiji Suzuki, and Takafumi Kanamori. *Density ratio estimation in machine learning*. Cambridge University Press, 2012\.
> > \[5\] Kanamori, Takafumi, Shohei Hido, and Masashi Sugiyama. "A least-squares approach to direct importance estimation." The Journal of Machine Learning Research 10 (2009): 1391-1445.

---

> > > ### Comment · Reviewer_ZnoF · 2025-11-21
> > >
> > > I thank the authors for the comprehensive and convincing response. I will raise my score if the authors make two minor changes (this will not take more than ten minutes):
> > >
> > > 1. I see that [2] is now referenced in the related work, as I suggested. However, [1] is still not referenced in the related work, only in the introduction. However, [1] clearly is related work, so it should be referenced again in the related work section. I do not understand why doing that poses a problem.
> > >
> > > 2. I find the answer in 1.1 convincing (isolation of the estimation problem into a testable scalar). However, I cannot see that this discussion is included in the updated manuscript. If it is and I just did not see it, please give me a pointer to where it is. Otherwise, I would be happy if the authors could add this discussion section to the main text of their manuscript.
> > >
> > > [1] Kladny, Klaus-Rudolf, Bernhard Schölkopf, and Michael Muehlebach. "Conformal Generative Modeling with Improved Sample Efficiency through Sequential Greedy Filtering." International Conference on Learning Representations (2025).
> > >
> > > [2] Shahrokhi, Hooman, et al. "Conformal prediction sets for deep generative models via reduction to conformal regression." Uncertainty in Artificial Intelligence (2025).

---

> > > > ### Author Response · Authors · 2025-11-21
> > > >
> > > > We sincerely thank the reviewer again for the timely feedback and for acknowledging our response. We have updated the manuscript to reflect the two changes:
> > > >
> > > > - The reference [1] is now added to Related works; we apologize for this oversight as we focused on adding the missing reference [2] in our previous revision.
> > > > - We have incorporated the discussion in point 1.1 to the manuscript, specifically,
> > > >   - the necessity of dealing with distribution shift was mentioned in the beginning of the new Section 3.3.
> > > >   - the first point that generative models do not come with natural built-in notion of reliability and ConfHit isolates the problem into a testable scalar was not incorporated in our last revision due to space limitations. Thank you for pointing it out, and we have now added this point to the end of Section 3.3 (we found it especially suitable after the discussion on robustness check).
> > > >
> > > > Please kindly see the revised manuscript for these changes. We would be more than happy to address any further comments!

---

> > > > > ### Comment · Reviewer_ZnoF · 2025-11-21
> > > > >
> > > > > I thank the authors for including the missing reference and pointing out the corresponding section that I could not find. I raised my score to *8: accept, good paper (poster)*. I wish the authors best of luck with this work.

---

### Official Review · Reviewer_PH3i · 2025-10-28

**Soundness:** 3
**Presentation:** 3
**Contribution:** 3
**Rating:** 4
**Confidence:** 3

**Summary:**

CONFHIT is a wrapper you put around any generator so that a small shortlist of molecules comes with a statistical promise: with confidence , at least one on this list is a hit. It does this without calling an oracle and while correcting for distribution shift between old data  new generations made from the generator.

**Strengths:**

Main strength is that this paper tries to tackle one of the critical problems in drug discovery. The fact that it is model-agnostic and no oracle is required is another strength. This makes the method actually usable in resource-constrained settings. THe Nested Testing Framework seems useful because it doesn't just certify N samples but returns the smallest certified subset

**Weaknesses:**

1. Coverage hinges on density-ratio correction (and the covariate-shift assumption).

2. May return an empty shortlist under tight α or small N. The paper notes that empty sets are inevitable without strong assumptions when the generation budget is limited, and empirically Bonferroni is almost always empty while CONFHIT still has ~16% empty sets in SBDD—improving on baselines but still a practical failure mode.

3. Needs an independently trained scoring model; power hinges on that predictor. The method seems to be only valid if the conformity/score function V is trained independently of the calibration/test data

**Questions:**

1. How robust is coverage when the density-ratio w(x) is misspecified—can you bound degradation or provide diagnostics a practitioner can run before trusting certification?

2. Can you offer principled guidance for choosing N and \alpha (or adaptive rules) that minimize empty-set rates while preserving guarantees?

3. How sensitive is CONFHIT to mild leakage or correlated training (e.g., feature reuse) between mu hat and calibration/test data, and can cross-fitting or data-splitting strategies be recommended to preserve coverage without sacrificing too much power?

---

> ### Author Response · Authors · 2025-11-21
> **Author response**
>
> We thank the reviewer for their constructive and thoughtful comments. We are encouraged that you found our paper well-presented, that the model-agnostic nature and “no-oracle” design are strengths, and that you recognized the practical utility of our nested testing framework. Below we address your concerns in detail.
>
> 1. **Robustness to misspecified density ratio estimation and diagnosis**
>    We agree that coverage depends on the quality of the estimated density ratio, and respond to your concern via (i) theory that bounds the degradation, (ii) empirical robustness, (iii) three new diagnosis methods.  A more detailed analysis of of these approaches is in response to Reviewer VPnq
>     1. **Bound on degradation**. The theoretical bound (Theorem 3.5) ensures that coverage degradation under misspecified w(x) is controlled by the magnitude of the estimation error.
>     2. **Empirical robustness**. Our empirical error control testifies the robustness of ConfHit to estimated density ratios. In addition, we now include new sensitivity experiments (Appendix H.1) where we perturb the estimated w(x). The results show that ConfHit maintains near-nominal coverage even with moderate mis-specification.
>     3. **Three new diagnosis methods**. We introduce simple diagnostics practitioners can run before certification, including (1) a *balance check* comparing calibration and test statistics, (2) validation shift approach, and (3) a *sensitivity probe* to examine the worst-case results under certain perturbations in w(x). These approaches are discussed in the new Section 3.3, with new experiments in Appendix H and discussion on the results in Section 4.5.
> 2. **Empty sets and choice of parameters.**
>    The parameter $\\alpha$ controls the error rate, and $N$ depends on the budget (how many samples the user is willing to generate in total). Larger $N$ or slightly relaxed $\\alpha$ naturally reduce the risk of empty sets while maintaining finite-sample guarantees. We would recommend users to choose these two values based on their goals and constraints.
>    We also note that occasional empty sets are not necessarily a failure mode: they indicate the model cannot confidently guarantee a hit under the specified budget. In this sense, the outcome *faithfully reflects the model’s uncertainty* rather than overstating its reliability. In practice, users can increase N or strengthen the generator to reduce this occurrence. Developing more powerful variants of ConfHit to further mitigate this remains an interesting direction for future work.
> 3. **Data leakage and data reuse in training.**
>    Thank you for highlighting this. We would like to first confirm that our training and calibration procedures follow standard conformal inference protocol: the scoring model (predictor) is trained independently from calibration data. Indeed, we want our method to work with any model the users find appropriate, especially pre-trained ones; and this is the case in our experiments.
>    We also appreciate the reviewer’s forward-looking question. Due to the unique structure of our problem, it is indeed possible to involve calibration and test data in training. The idea is to impute “negative” labels for the test data and use all the data to train the model. We can show that as long as the algorithm is not sensitive to the ordering of the data, our theoretical guarantee still goes through. We include an extended version of ConfHit in Appendix A.6 with a pointer and discussion at the end of Section 3.1.
>
>
> **Summary**: We have strengthened the robustness results and provided new diagnostics, adaptive parameter guidance, and expanded documentation to make the method easier to apply in practice. We hope these clarifications demonstrate that ConfHit is both *theoretically principled* and *practically robust*, addressing key limitations that have prevented conformal reliability from being usable in real-world generative settings.

---

### Official Review · Reviewer_SYBt · 2025-10-29

**Soundness:** 3
**Presentation:** 3
**Contribution:** 2
**Rating:** 6
**Confidence:** 2

**Summary:**

The article describes variant of conformal prediction for guaranteeing that a batch of samples generated from a generative model satisfies a predefined property. The use case in mind is molecular generation, where such a predefined property could be a valid base-scaffold or a certain amount of improvement over an existing seed molecule.

**Strengths:**

The article is overall well-written and the concepts are presented in a clear and coherent manner. Numerical examples indicate that the method can indeed be applied to molecular generation problems.

**Weaknesses:**

- The core contribution is rather incremental from a statistical/mathematical perspective. The ConfHit algorithm (Alg. 1) is a variant of a permutation test that has been known in the literature for a long time. The idea of incorporating nested testing is a relatively straightforward addition (and I would be surprised if similar techniques have not been used before, maybe in a slightly different context).

- The significance of the numerical results are difficult to interpret (I am also not an expert in molecular generation). However, baseline comparisions against earlier methods are omitted with the argument that a completely new set-up is studied, where there is a distribution shift between the calibration data and the distribution of the generative model. Nonetheless, in the absence of such distribution shift a comparison to the relevant baselines Quach et al., 2023, and Kladny et al., 2024 seems doable and provides an indication of the size and the quality of the generated set.

- The method requires the estimation of a covariate shift w between the distribution used for calibration and the distribution of the generated samples. It might be difficult to reliably estimate w in practice.

- The latex-style file has been modified to ommit the label Figure 5, possibly to deal with page restrictions. I recommend the authors not to do this in future submissions, as it is an unfair practice and seems to be against the submission rules.

- The guarantees hold marginal and not hold conditional on the calibration data. This could be emphasized and explained more as it might otherwise mislead and misguide the interpretation of the statistical guarantee. Moreover, for getting a guarantee conditional on the calibration data, one would potentially need a large calibration dataset, exacerbating the computational challenge (permutation testing) of the proposed approach.

**Questions:**

- ConfHit is based on a permutation test, where permutations are subsampled to tame the computational effort. However, the subsampling introduces variance that only decreases with 1/B. Hence estimation of p-values might be difficult, in particular if these are small and might require large B and significant computational effort. Have the authors experienced issues along these lines? In order words, would one be able to obtain similar results if computational time is reduced from 30min (App. C3) to 5min.

---

> ### Comment · Reviewer_ZnoF · 2025-11-13
>
> I would like to comment on the point *"Nonetheless, in the absence of such distribution shift a comparison to the relevant baselines Quach et al., 2023, and Kladny et al., 2024 seems doable and provides an indication of the size and the quality of the generated set."*. I agree with the authors here that a comparison with these methods is not fair, because the proposed method does not assume access to an oracle. It would surely be a nice-to-have for the appendix (it would answer the question: "how much does the oracle even help?"), but certainly it is not necessary for the main text.

---

> ### Author Response · Authors · 2025-11-21
> **Author response 1**
>
> We thank the reviewer for their thoughtful and detailed comments, and we appreciate that you found the paper clearly written, conceptually coherent, and the numerical experiments informative. We address each of your points below.
>
> 1. **Novelty and contributions**
>    We appreciate your observation that ConfHit draws from classical ideas such as permutation testing and multiple hypothesis testing. We fully agree, and would like to clarify how our contribution lies in *restructuring, advancing and unifying these techniques* to address a fundamentally new setting: **finite-sample, distribution-shifted generative design**. Our specific techniques are new, and this combination of goals has not been addressed by prior conformal or permutation-based frameworks.
>     a. **Weighted permutation test and new statistical structure**.  While permutation-based inference is classical, **ConfHit introduces a new structure that integrates density-ratio-weighted calibration with multi-point conformal testing under covariate shift**. This enables novel finite-sample validity in drug discovery problems. Technically, we would like to respectfully note that permutation testing is the conceptual foundation of many modern inferential methods, including conformal prediction, and continued advances built on this foundation remain highly valuable. In particular, the **covariate-shift extension of permutation test** (weighted two-sample tests) was not studied until very recently, and the literature is still growing, which our method contributes to \[1\]. Therefore, we would like to argue the statistical novelty of ConfHit with significant practical relevance.
>
>     b. **Novelty of nested testing**. The second new element of ConfHit is nested testing, which allows simultaneous assessment of multiple batches without the need of adjustment for multiplicity. We acknowledge its conceptual adjacency to classical ideas such as Holm’s test and closed testing in general. However, instead of a naive application of existing methods, it operates across generated batches with multi-sample weighted permutation test to unlock the new capabilities of the design problem. We highlight that this structure is not a straightforward reuse of existing statistical tools; it reinterprets multiplicity correction for a constrained generative process, leading to both theoretical and practical advances.
>    We hope this clarifies that while ConfHit is inspired by prior ideas, it integrates them in a new way to solve a practically important problem that has not previously been addressed. We would be grateful if the reviewer could point out any specific aspects they feel remain too close to existing work so that we can address them more precisely.
> 2. **Baseline comparison**
>    We thank the reviewer for highlighting this and sincerely appreciate the thoughtful follow-up discussion. Following your suggestion, we have added two additional baselines for comparison.
>     * First, we include a heuristic baseline that uses the model-produced probabilities to determine a stopping point; as discussed in Appendix G.1, this method is highly sensitive to predictor noise and fails to provide reliable coverage. In contrast, ConfHit is model-agnostic and maintains validity under the same model degradation (Appendix F). These results show that principled uncertainty quantification with proper density ratio estimation provides robust guarantees.
>     * Second, we evaluate a conformal LM–style baseline that uses an (unavailable in practice) oracle predictor and performs calibration without any density correction in G2. Even under this assumption, the method breaks coverage and underperforms our approach.

---

> > ### Author Response · Authors · 2025-11-21
> > **Author response 2**
> >
> > 3. **Significance of results in molecule generation/drug discovery \-**
> >    Thank you for raising this concern. Both constrained molecule generation and structure-based drug discovery are widely studied and practically important problems in modern ML, with substantial recent work including JT-VAE \[2\], Chemprop/MPNNs \[3\], DiffDock \[4\] and TargetDiff \[5\]. These tasks also mirror real drug-discovery workflows, where hit identification and lead optimization are routine but extremely expensive components of the pipeline; experimental validation campaigns often cost hundreds of millions of dollars, representing a major bottleneck in early-stage R\&D \[6, 7\]. Because of these high costs, having robustness guarantees on generative models is highly valuable: certifying that a generated set contains a viable candidate before any wet-lab work can substantially reduce experimental burden and decision-making uncertainty. As far as we know, no prior work provides such reliability guarantees for generative models in molecular design, and oracle-based calibration is impractical because it would require executing wet-lab assays. Our numerical results therefore demonstrate that ConfHit offers meaningful, distribution-free guarantees in settings which have not previously been available.
> >
> >
> > 4. **Estimation of density ratio.**
> >    We agree that accurate estimation of the density ratio is challenging and crucial. We clarify this point and update the manuscript from several angles: (i) robustness of our methods, (ii) three new diagnosis methods, and (iii) rich toolboxes for density ratio estimation. These approaches are discussed in the **newly added Section 3.3, with new experiments in Appendix H and discussions in Section 4.5.** A more detailed analysis of of these approaches is in response to Reviewer VPnq
> >     1. We note several newly added results to show the robustness of ConfHit to estimation error. Besides (i) theory that bounds the degradation, we now added (ii) empirical robustness, and (iii) three new diagnosis methods.
> >        1. **Empirical robustness**. Our empirical error control testifies the robustness of ConfHit to estimated density ratios. We include new sensitivity experiments (Appendix H.1) where we perturb the estimated w(x). ConfHit maintains near-nominal coverage even with moderate mis-specification.
> >        2. **Three new diagnosis methods**. We introduce simple diagnostics practitioners can run before certification, including (1) a *balance check* comparing calibration and test statistics, (2) validation shift approach, and (3) a *sensitivity probe* to examine the worst-case results under certain perturbations in w(x). These approaches are discussed in the new Section 3.3, with new experiments in Appendix H and discussion on the results in Section 4.5.
> >    2. Estimating the density ratio is a statistically standard problem used in many fields like importance weighting, covariate shift, and causal inference. In practice, the ratio can be learned using kernel or classification-based methods that are empirically robust even in high dimensions (instead of estimating two separate densities). We have clarified our implementation, noting that we use kernel density estimation following the success of prior work (CoDrug), and that alternative estimators (e.g., logistic regression-based likelihood ratio estimation) can also be employed effectively. As density ratio estimation is widely used in importance adjustment, covariate shift, and causal inference, there is a rich toolbox we can borrow from these mature literatures. We thoroughly discussed these points in the newly added Section 3.3.
> > 5. **Figure 5 caption**
> >    We apologize for the missing label in Figure 5\. This was an unintentional mistake: we used separate captions for subfigures, and these are all the information we intended to include, but we didn’t realize the main label is gone. We have fixed this in the updated manuscript.
> > 6. **Calibration-conditional guarantees.**
> >    Thank you for this careful observation. We agree that our guarantees hold *marginally* rather than *conditionally* on calibration data, and we have made this explicit in the revised text. In practice, conditioning will change the statistical structure of the problem (we cannot leverage the exchangeability any more, but may need to estimate the success probability, which is very challenging.)

---

> > > ### Author Response · Authors · 2025-11-21
> > > **Author response 3**
> > >
> > > 7. **Computation efficiency.**
> > >    Thank you for raising this point. We have clarified the computation cost by explicitly separating the runtime for molecule generation from the cost of running ConfHit in Appendix C.4. The dominant computational cost comes from generating candidate molecules, which is inherent to all molecular optimization pipelines and not specific to our method. The ConfHit procedure itself is lightweight: the full conformal step takes only about 4 minutes for 800 samples for B=2000 (roughly 0.3 seconds per sample). We also evaluate the impact of different permutation budgets, ranging from 500 to 5000 and find no meaningful change in empirical variance or performance, with runtime scaling nearly linearly. Given this modest computational burden, we recommend using a budget of around 1000 permutations, which provides a reliable calibration layer with very little additional computation.
> > >
> > >
> > >
> > > **Summary:** We sincerely appreciate the reviewer’s engagement and insightful comments, which have helped us strengthen the paper. We have clarified the theoretical distinctions from prior work, expanded empirical baselines, added robustness analyses, fixed formatting issues, and clarified on the computation costs. We hope these revisions help convey the novelty and practical value of ConfHit more clearly, and would be more than happy to address any more concerns the reviewer might have.
> > >
> > > \[1\] Hu, Xiaoyu, and Jing Lei. "A two-sample conditional distribution test using conformal prediction and weighted rank sum." Journal of the American Statistical Association 119.546 (2024): 1136-1154.
> > > \[2\] Jin et al., “Junction Tree Variational Autoencoder for Molecular Graph Generation,” ICML 2018\.
> > > \[3\] Yang et al., “Analyzing Learned Molecular Representations for Property Prediction,” JCIM 2019\.
> > > \[4\] Corso et al., “DiffDock: Diffusion Steps, Twists, and Turns for Molecular Docking,” ICLR 2023\.
> > > \[5\] Gligorijević et al., “TargetDiff: Diffusion Models for Structure-Based Drug Design,” 2024\.
> > > \[6\] Paul et al., “How to improve R\&D productivity: the pharmaceutical industry’s grand challenge,” Nat. Rev. Drug Discov. 2010\.
> > > \[7\] DiMasi et al., “Innovation in the pharmaceutical industry: New estimates of R\&D costs,” J. Health Econ. 2016\.

---

> > > > ### Comment · Reviewer_SYBt · 2025-11-26
> > > > **Rebuttal Acknowledgment**
> > > >
> > > > Dear authors,
> > > >
> > > > I would like to thank you for the time and effort spent on revising the manuscript, and answering my questions. The additions clearly strengthen the manuscript. I am satisfied with the changes/answers and will argue for acceptance in the subsequent discussion period.

---

### Official Review · Reviewer_VPnq · 2025-11-01

**Soundness:** 3
**Presentation:** 3
**Contribution:** 3
**Rating:** 6
**Confidence:** 3

**Summary:**

The paper introduces CONFHIT, a framework that brings formal reliability guarantees to conditional generative design under tight experimental budgets. Instead of assuming an oracle that can instantly validate new samples, the method leverages historical labeled data and corrects distribution shift between past and newly generated candidates through density ratio weighting. It builds joint conformal p values over batches to certify that a generated set contains at least one viable hit and extends this to a nested testing strategy that prunes a large batch to a compact shortlist while retaining finite sample error control. The authors ground the approach with theory and evaluate it on constrained molecule optimization and structure based drug discovery, showing tight error control across target levels and models, practical reductions in certified set size, and sensible behavior in ablations that remove density correction.

**Strengths:**

The problem is well framed around the realities of discovery work where assay budgets are limited and oracle access is infeasible, and the split between certification and design clarifies when guarantees are possible and how to act when they are. The use of weighted exchangeability to address covariate shift is principled and connects cleanly to modern conformal theory, and the nested testing idea is simple to implement yet surprisingly powerful, needing only monotone individually valid p values to avoid multiplicity penalties. The empirical study spans multiple model classes and tasks and reports both coverage and power with clear plots, while the ablation on removing density correction provides convincing evidence that the weighting is not cosmetic. The paper is mostly model agnostic in how it defines conformity scores and offers several reasonable choices, which improves the portability of the framework. The budget allocation vignette is a useful bridge from statistics to program level decision making.

**Weaknesses:**

The guarantees depend critically on accurate density ratio estimates built on learned features, and the robustness analysis, while helpful, still leaves open how to diagnose and mitigate misspecification in practice when calibration data are scarce or shift is large. The approach relies on a property predictor to define the conformity score, so end to end performance will be sensitive to that model and feature extractor, yet guidance on model selection, calibration, and failure modes is limited. Comparisons to prior conformal generative methods are discussed as not directly applicable because of oracle assumptions, but the experimental section would be stronger with alternative practical baselines beyond Bonferroni, for example heuristic pruning driven by predictor uncertainty, or variants that reuse unlabeled generated data for semi supervised calibration. Computational cost may be nontrivial due to permutation sampling and kernel density estimation, and the paper does not provide a thorough accounting of wall clock cost versus gains in certified set size. The empirical scope focuses on small molecule scenarios with in silico oracles rather than wet lab measurements, so external validity to other domains such as proteins or materials remains uncertain. Finally, some design choices are only briefly justified, such as training the density model on a subset and the specific feature layers used, and readers may benefit from clearer guidance on hyperparameters like the number of permutations, batch budgets, and how to pick among score statistics without post hoc tuning.


---


Itemized weaknesses for rebuttal and discussion:

1. Heavy reliance on accurate density ratio estimates with limited guidance for diagnosing misspecification
2. Sensitivity to the property predictor and feature extractor with sparse advice on model selection and calibration
3. Limited baselines beyond Bonferroni; lacks comparisons to stronger practical heuristics or semi supervised variants
4. Computational overhead from permutation testing and density modeling without clear accounting of wall clock cost
5. Evaluation focused on in silico small molecule settings, leaving external validity to other domains uncertain
6. Several design and hyperparameter choices are under explained, reducing reproducibility and practitioner guidance

**Questions:**

-

---

> ### Author Response · Authors · 2025-11-21
> **Author response 1**
>
> We thank the reviewer for their thoughtful feedback and constructive suggestions. We appreciate that you found the problem well-motivated, our method principled and powerful, and our experiments realistic and comprehensive. Below we address each main point raised.
>
> 1. **Reliance on density ratio estimates and guidance for diagnosis**
>    We fully agree on the importance of density ratio estimation in our framework. The difference between historical assay data and newly generated molecules is a long-standing issue in this field and in nearly all conformal prediction-related works \[1,2,3\]. In the meantime, we recognize the limited scope of our initial robustness analysis and have expanded our discussion in several aspects in the **newly added Section 3.3, with new experiments in Appendix H and discussions in Section 4.5**:
>    1. **Robustness**: To study the impact of misspecification, we adversarially perturb the fitted density ratios by a power law scaling and examine the resulting error rates of ConfHit. Our new results are in Appendix H which reveal a reasonable degradation of the error rate (discussion in Section 4.5).
>    2. **Diagnosis 1: balance check**. We added a discussion on diagnosis via balance check. A correct density ratio should make the sample mean of any function of X to be close between calibration and test samples. Following this principle, we discuss the balance check as a diagnosis in Section 3.3 and empirically examine the balance in our experiments to provide example usage in Appendix H.2 with discussion in Section 4.5.
>    3. **Diagnosis 2: validation shift**. The second approach is to split the labeled data in a non-uniform way to create nontrivial synthetic distribution shift, and run ConfHit to validate its ability to correct for it by checking the uniformity of p-values on inactive validation samples. An example in our experiments is in Appendix H.3 and discussed in Section 4.5.
>    4. **Diagnosis 3: sensitivity analysis**. Finally, inspired by sensitivity analysis, we propose to probe the worst-case result under misspecification. The idea is to assume the true weights are within a range around the fitted values, and check the worst-case testing among these scenarios. An example of this kind is used in our experiments to examine robustness; see Appendix H.1 and discussion in Section 4.5.
>
> 2. **Sensitivity to property predictor and feature extractor**
>    We appreciate this concern. ConfHit is designed to be model-agnostic as other methods in conformal inference, and its validity does not rely on the property predictor. Practitioners can thus freely use the most suitable prediction model in the application, or choose the most powerful model available (in a way that is independent of the calibration and test data, e.g., on a holdout “training” fold).
>    In the case a less powerful property predictor (conformity score) is used, ConfHit will be less powerful in identifying hits, leading to larger certified sets yet validity is preserved. This point has been discussed at the end of Section 3.1 (where we talked about the score choice). To investigate this point, we have also added new experiments in Appendix F with a weak property predictor by injecting random noise to the predicted values, and discuss the results and findings in Section 4.5.
> 3. **Limited baselines beyond Bonferroni**
>    We thank the reviewer for this valuable suggestion. While directly comparable methods are not available, given the novel problem setup and unique constraints in our setting, we have additionally examined **two baselines** in Appendix G.
>    In Section G.1, we introduce a heuristic baseline that mimics a likelihood-ratio-based stopping rule that uses the estimated likelihood ratios to determine when to stop sampling. This baseline depends heavily on the quality of the predictive model. We observe that its error control is highly sensitive to predictor noise and is frequently violated when the likelihood estimates are imperfect. In stark contrast, the error control of ConfHit is agnostic to predictor quality which we illustrate in Appendix F in response point 2 above.
>    Further, we add an **oracle-based baseline** that applies a conformal language-modeling style procedure with oracles but without any density correction. Even under this unrealistic oracle assumption, the method fails to maintain valid coverage and exhibits lower power compared to our approach.
>    These new experiments highlight that both heuristic and oracle baselines struggle in the presence of distribution shift, whereas our proposed method achieves reliable guarantees and without the need of oracle.

---

> > ### Author Response · Authors · 2025-11-21
> > **Author response 2**
> >
> > 4. **Computation overhead**
> >    We appreciate this important point and clarify that the dominant computational cost in our pipeline arises from candidate generation, which is inherent to all molecular generative modeling workflows rather than ConfHit. We have included an updated runtime breakdown in Appendix C.4, and observe that the majority of runtime comes from model training and sample generation (inherent in the tasks). The ConfHit procedure itself is lightweight: running the full conformal certification step requires only 4 minutes for 800 test samples (≈0.30 seconds/sample) of the 30 minutes. Thus, ConfHit provides robust distribution-free guarantees with negligible additional computational overhead on top of existing generative pipelines.
> >
> >    **Advantage over other conformal generative methods**. As it does not require any more generation beyond what is already in the generative task, this further highlights the advantage of ConfHit over other conformal generative methods (which need to  generate new samples for “calibration” inputs and can be very costly \[4,5,6\]).
> >
> > 5. **Evaluation limited on in silico small molecule settings**
> >    We fully acknowledge this point and have discussed this in the paper. We emphasize that small molecules are currently the most well-studied and well-resourced domain for conditional generative inference (such as pocket-conditioned generation or lead optimization), with widely available benchmarks, docking models, and predictive scoring functions, which makes them a suitable and realistic testbed for evaluating reliability. Other fields like protein generation are also practically important, but remain bottlenecked by limited ground-truth data, weaker predictive oracles and performing conformal calibration without a reliable predictor is currently infeasible. We view the extension of our framework to larger biological systems as a promising future direction
> > 1. **Design and hyperparameter choices**
> >    Thank you for raising this issue. We have expanded our discussion of hyperparameters and design choices in Appendix C.2, providing clearer justification and guidance for all components of the method. To further ensure reproducibility, we will release the full code repository publicly upon acceptance. The calibration code, sampled datasets, and predictor models are already included with the submitted supplementary materials. Please let us know if additional details would be helpful, we are happy to incorporate any further clarifications the reviewer may suggest.
> >
> > **Summary:** We appreciate the reviewer’s thoughtful feedback and positive assessment of our theory and experiments. We have updated the paper with new robustness studies, expanded diagnostics, additional baselines, detailed runtime analysis, and detailed discussion on design choices.
> >
> > \[1\] Krstajic, Damjan. "Critical assessment of conformal prediction methods applied in binary classification settings." Journal of Chemical Information and Modeling 61.10 (2021): 4823-4826.
> > \[2\] Fannjiang, Clara, and Ji Won Park. "Reliable algorithm selection for machine learning-guided design." arXiv preprint arXiv:2503.20767 (2025).
> > \[3\] Laghuvarapu, Siddhartha, Zhen Lin, and Jimeng Sun. "Codrug: Conformal drug property prediction with density estimation under covariate shift." Advances in Neural Information Processing Systems 36 (2023): 37728-37747.
> > \[4\] Victor Quach, Adam Fisch, Tal Schuster, Adam Yala, Jae Ho Sohn, Tommi S Jaakkola, and Regina Barzilay. Conformal language modeling. *arXiv preprint arXiv:2306.10193*, 2023\.
> > \[5\] Klaus-Rudolf Kladny, Bernhard Schölkopf, and Michael Muehlebach. Conformal generative modeling with improved sample efficiency through sequential greedy filtering. *arXiv preprint arXiv:2410.01660*, 2024\.
> > \[6\] Shahrokhi, Hooman, et al. "Conformal prediction sets for deep generative models via reduction to conformal regression." Uncertainty in Artificial Intelligence (2025).

---

### Author Response · Authors · 2025-11-21
**Summary of Revisions and Responses to Reviewers**

We sincerely thank all reviewers for their constructive and insightful feedback. We appreciate the reviewers' recognition of the clarity of the writing and notation (SYBt, PH3i, ZnoF), the strong and practically grounded problem formulation (VPnq, ZnoF), and the principled integration of weighted exchangeability and nested testing for reliable certification (VPnq). Reviewers further noted the simplicity and model-agnostic nature of the approach, as well as its practicality in resource-limited settings without reliance on an oracle (PH3i, VPnq), together with the strength of the theoretical analysis and empirical evaluation (VPnq, ZnoF).

Reviewer SYBt and Reviewer ZnoF have participated in the subsequent discussion and both have confirmed that their concerns were fully addressed. [Reviewer SYBt](https://openreview.net/forum?id=IruPup3KnX&noteId=EMBJTk2XOv) noted that **“the additions clearly strengthen the manuscript”** and stated that they are **“satisfied with the changes/answers and will argue for acceptance.”** [Reviewer ZnoF](https://openreview.net/forum?id=IruPup3KnX&noteId=vio2Uzmjkk) updated their score, describing the rebuttal as a **“comprehensive and convincing response”** and writing, **“I raised my score to 8: accept, good paper (poster).”**

Below, we summarize the substantial improvements to the paper based on the reviewers’ suggestions. Changes are marked in red in the manuscript and summarized below:

* In response to requests for additional robustness guidance/diagnostics for density ratio estimation (Reviewers VPnq, SYBt, PH3i, ZnoF):
    * Updated Section 3.4 to provide an expanded discussion on the robustness of density-ratio estimation and added three diagnostics for estimation quality.
    * New experiments analyzing density-ratio robustness and estimation quality, including:
        * Sensitivity analysis under adversarial perturbations of weights (Appendix H.1),
        * Balance check measuring alignment between calibration and test distributions (Appendix H.2),
        * Synthetic scaffold-split experiment mimicking realistic drug-discovery shifts (Appendix H.3).
    * These analyses are synthesized and discussed in the new Section 4.5 “Robustness check.”

* To enable data reuse for training and calibration/test (Reviewer PH3i):
    * Introduced a new variant of ConfHit without sample splitting (Appendix A.6, referenced in Section 3.1), designed for data-limited settings.

* Additional baselines to compare with unguaranteed heuristic procedures and oracle-based methods (Reviewer VPnq, SYBt):
    1. A heuristic baseline using predicted probabilities (Appendix G.1), illustrating the necessity of principled uncertainty quantification for robust guarantees.
    2. An oracle baseline using true labels without density correction (Appendix G.2), demonstrating that shift-adjusted calibration is essential to simultaneously achieve error control and strong power.

* Provided clarification of run-time analysis (Reviewer VPnq, SYBt):
    * Clarified running-time analysis (Appendix C.4), showing that ConfHit adds only minimal overhead relative to the inherent cost of molecule generation.

* Provided additional information about design choices and implementation details to further enhance reproducibility (Reviewer VPnq):
    * Expanded discussion of hyperparameters and implementation details (Appendix C.2), including density-ratio estimation choices, predictive model selection, test-statistic behavior, and permutation-budget benchmarks.

* Sensitivity to property predictor/effect when a weak property predictor is used (Reviewer VPnq):
    * Added new experiments in Appendix F with a weak property predictor by injecting random noise into the predicted values, and discussed the results and findings in Section 4.5.

* Clarified novelty, contributions, and significance of the results in molecule/drug discovery in the response(Reviewer SYBt).

---

### Meta-Review · Area_Chair_uouj · 2026-01-08

**Summary:**

ConfHit addresses a critical gap in AI-driven scientific discovery: the lack of formal guarantees that generated candidates (like molecules) satisfy desired properties. In real-world drug discovery, testing every generated molecule in a "wet lab" (the oracle) is prohibitively expensive. ConfHit provides a way to certify results using only historical data and statistical inference.

The reviewers generally praised the paper’s clarity and the practical importance of the problem. However, they raised several technical and "significance" concerns, most of which were addressed in the revised manuscript.

I agree with the reviewers.

**Reviewer Concerns:**

Reviewers (VPnq, SYBt, ZnoF) argued that the framework’s validity depends heavily on the accuracy of the density ratio $w(x)$. If the estimation of the shift is wrong, the guarantees might fail.

Reviewer SYBt suggested the math (permutation tests and nested testing) was a standard statistical toolset applied to a new area.

**Reviewer Scores:**

Reviewer SYBt and Reviewer ZnoF have participated in the subsequent discussion and both have confirmed that their concerns were fully addressed. Reviewer SYBt noted that “the additions clearly strengthen the manuscript” and stated that they are “satisfied with the changes/answers and will argue for acceptance.” Reviewer ZnoF updated their score, describing the rebuttal as a “comprehensive and convincing response” and writing, “I raised my score to 8: accept, good paper (poster).”

---

### Decision · Program_Chairs · 2026-01-26

Accept (Poster)